# FedGT: Identification of Malicious Clients in Federated Learning with Secure Aggregation

## Abstract

We propose FedGT, a novel framework for identifying malicious clients in federated learning with secure aggregation. Inspired by group testing, the framework leverages overlapping groups of clients to identify the presence of malicious clients in the groups via a decoding operation. The clients identified as malicious are then removed from the training of the model, which is performed over the remaining clients. By choosing the size, number, and overlap between groups, FedGT strikes a balance between privacy and security. Specifically, the server learns the aggregated model of the clients in each group—vanilla federated learning and secure aggregation correspond to the extreme cases of FedGT with group size equal to one and the total number of clients, respectively. The effectiveness of FedGT is demonstrated through extensive experiments on the MNIST, CIFAR-10, and ISIC2019 datasets in a cross-silo setting under different data-poisoning attacks. These experiments showcase FedGT's ability to identify malicious clients, resulting in high model utility. We further show that FedGT significantly outperforms the private robust aggregation approach based on the geometric median recently proposed by Pillutla *et al.* on heterogeneous client data (ISIC2019) and in the presence of targeted attacks (CIFAR-10 and ISIC2019).

## 1 Introduction

Federated learning (FL) (McMahan et al., 2017) is a distributed machine learning paradigm that enables multiple devices (clients) to collaboratively train a machine learning model under the orchestration of a central server while preserving the privacy of their raw data. To preserve privacy, the clients share their local models instead of the raw data with the central server.

In its original form, FL is susceptible to model-inversion attacks (Fredrikson et al., 2015; Wang et al., 2019), which allow the central server to infer clients' data from their local model updates. As demonstrated in (Dimitrov et al., 2022), such attacks can be mitigated by employing secure aggregation protocols (Bonawitz et al., 2017; Bell et al., 2020). These protocols guarantee that the server only observes the aggregate of the client models instead of individual models.

A salient problem in FL is poisoning attacks (Baruch et al., 2019), where malicious and/or faulty clients corrupt the jointly-trained global model by introducing mislabeled training data (*data poisoning*) (Tolpegin et al., 2020; Wang et al., 2020), or by modifying local model updates (*model poisoning*) (Fung et al., 2018). Poisoning attacks pose a serious security risk for critical applications. Defensive measures against these threats generally fall into two categories: robust aggregation and anomaly detection. Robust aggregation techniques (Blanchard et al., 2017; Yin et al., 2018; Cao et al., 2019) are reactive approaches designed to mitigate the effect of poisoned models, whereas anomaly detection is inherently proactive and aims to identify and eliminate corrupted models (Li et al., 2020; Mallah et al., 2021; Nguyen et al., 2022). Robust aggregation techniques can introduce bias, especially when clients have heterogeneous data Li et al. (2020), and their effectiveness tends to diminish with an increasing number of malicious clients Zhang et al. (2022). Moreover, a recurring issue with defense mechanisms is their reliance on accessing individual client models, leaving clients vulnerable to model-inversion attacks. Addressing resiliency against poisoning attacks and devising protocols for the identification of malicious clients in FL without access to individual client models remains a challenge (Kairouz et al., 2021; Gong et al., 2023). Notably, privacy-enhancing schemes such as

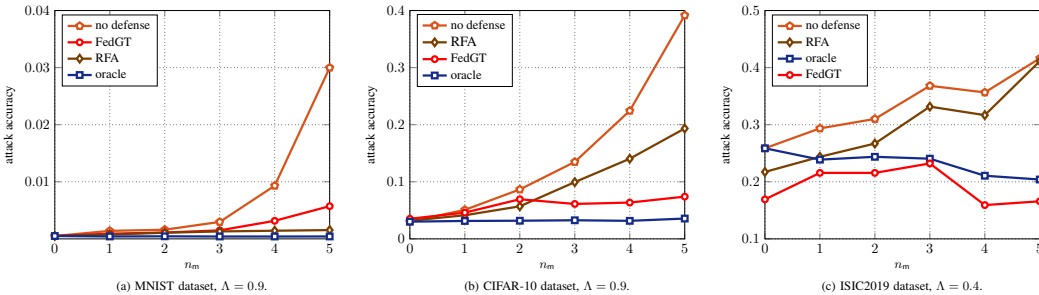

Figure 1: Attack accuracy MNIST (left), CIFAR10 (mid), and ISIC2019 (right) datasets for varying number of malicious clients.

secure aggregation, enhance clients' privacy at the expense of camouflaging adversaries (Kairouz et al., 2021). Hence, there is a fundamental trade-off between security and privacy.

In this paper, we propose FedGT, a novel framework for identifying malicious clients in FL with secure aggregation. Our framework is inspired by group testing (Dorfman, 1943), a paradigm to identify defective items in a large population that significantly reduces the required number of tests compared to the naive approach of testing each item individually. FedGT's key idea is to group clients into overlapping groups. For each group, the central server observes the aggregated model of the clients and runs a suitable test to identify the presence of malicious clients in the group. The malicious clients are then identified through a decoding operation at the server, allowing for their removal from the training of the global model. FedGT trades-off client's data privacy, provided by secure aggregation, with *security*, understood here as the ability to identify malicious clients. It encompasses both non-private vanilla FL and privacy-oriented methods, e.g., secure aggregation, by selecting group sizes of one and the total amount of clients, respectively. However, by allowing group sizes between these two extremes, FedGT strikes a balance between privacy and security, i.e., improved identification capabilities comes at the cost of secure aggregation involving fewer clients.

We showcase FedGT's effectiveness in identifying malicious clients through experiments on the MNIST, CIFAR-10, and ISIC2019 datasets under both targeted and untargeted data-poisoning attacks. Our focus is specifically on the cross-silo scenario, wherein the number of clients is moderate (up to 100) and data-poisoning is the predominant attack vector (Shejwalkar et al., 2022). Fig. 1 illustrates the performance of FedGT for a scenario with 15 clients and a targeted label-flipping attack. The figure shows the attack accuracy after 10 (MNIST), 50 (CIFAR-10), and 40 (ISIC2019) communication rounds, respectively, versus the number of malicious clients, $n_{\mathsf{m}}$. When no defense mechanism is in place, the attack success significantly increases as the number of malicious clients grows. Remarkably, FedGT enables the identification and removal of malicious clients with low misdetection and false alarm probabilities. This leads to a substantially reduced attack accuracy—significantly outperforming the recently-proposed robust federated aggregation (RFA) protocol based on the geometric median (Pillutla et al., 2022) for the CIFAR-10 and ISIC2019 datasets, while achieving a lower communication complexity.

## 2 RELATED WORK

To the best of our knowledge, only the works (So et al., 2021; Pillutla et al., 2022; Zhang et al., 2021) address resiliency against poisoning attacks in conjunction with secure aggregation. The work by So et al. (2021) is the first single-server solution to account for both privacy and security in FL. The protocol is based on drop-out resilient secure aggregation where the server utilizes secret sharing to first obtain the pairwise Euclidean distance between the clients' updates and then selects what clients to aggregate by means of multi-Krum (Blanchard et al., 2017). However, it is not clear if the pairwise differences can leak extra information. In (Pillutla et al., 2022), a robust aggregation protocol (dubbed RFA) based on an approximate geometric median (computed exploiting secure aggregation) is proposed. However, this protocol lacks the capability to identify malicious clients and is known to be inferior to other robust aggregation techniques, especially when dealing with heterogeneous client data (Li et al., 2023). The work by Zhang et al. (2021) presents a privacy-preserving tree-based robust aggregation method. In particular, each leave in the tree consists of a subgroup of clients who securely aggregate their local models. To achieve privacy between subgroups, masking is done on all but the

last parameters in the aggregated models. By using the Euclidean distance between the unmasked parameters and the corresponding parameters in the global model, an outlier removal scheme, based on variance thresholding, is used iteratively to determine what groups should contribute to the global model. The approach in Zhang et al. (2021) is the method closest to ours as it relies on dividing clients into subgroups and on testing the group aggregates. However, contrary to FedGT, it is unable to identify malicious clients and to leverage the information of overlapping groups.

## 3 PRELIMINARIES

**Group testing.** Group testing (Dorfman, 1943; Aldridge et al., 2019) encompasses a family of test schemes aiming at identifying items affected by some particular condition, usually called *defective* items (e.g., individuals infected by a virus), among a large population of $n$ items (e.g., all individuals). The overarching goal of group testing is to design a testing scheme such that the number of tests needed to identify the defective items is minimized. The principle behind group testing is that, if the number of defective items is significantly smaller than $n$, then negative tests on groups (or pools) of items can spare many individual tests. Following this principle, items are grouped into overlapping groups, and tests are performed on each group. Based on the test results on the groups, the defective items can then be identified—in general with some probability of error—via a decoding operation.

**Attack model.** We consider a cross-silo scenario with an honest-but-curious server and $n$ clients out of which $n_{\mathsf{m}}$ are compromised (referred to as *malicious* clients). A client may be compromised due to hardware malfunction or adversarial corruption. In the latter case, we assume that the malicious clients can collude and perform coordinated attacks against the global model. In this paper, we focus on data-poisoning attacks, which constitute the most realistic type of attack for cross-silo FL (Shejwalkar et al., 2022).

## 4 FEDGT: GROUP TESTING FOR FL WITH SECURE AGGREGATION

We consider a population of $n$ clients, $n_{\mathsf{m}}$ of which are malicious. We define the *defective vector* $\boldsymbol{d} = (d_1, d_2, \ldots, d_n)$ with entries representing whether a client $j$ is malicious ($d_j = 1$) or not ($d_j = 0$). It follows that $\sum_{j=1}^{n} d_j = n_{\mathsf{m}}$. Note that $\boldsymbol{d}$ is unknown, i.e., we do not know a priori which clients are malicious.

Borrowing ideas from group testing (Dorfman, 1943), the $n$ clients are grouped into $m$ overlapping *test groups*. We denote by $\mathcal{P}_1, \mathcal{P}_2, \ldots, \mathcal{P}_m$ the set of indices of the clients belonging to test group $i$, i.e., if client $j$ is a member of test group $i$, then $j \in \mathcal{P}_i$.

**Definition 1** (Assignment matrix)**.** The assignment of clients to test groups can be described by an assignment matrix $\boldsymbol{A} = (a_{i,j})$, $i \in [m]$, $j \in [n]$, where $a_{i,j} = 1$ if client $j$ participates in test group $i$ and $a_{i,j} = 0$ otherwise.

The assignment of clients to test groups, i.e., matrix $\boldsymbol{A}$, can be conveniently represented by a bipartite graph consisting of $n$ *variable nodes* (VNs) $\mathsf{v}_1, \ldots, \mathsf{v}_n$ corresponding to the $n$ clients, and $m$ *constraint nodes* (CNs) $\mathsf{c}_1, \ldots, \mathsf{c}_m$, corresponding to the $m$ test groups. An edge between VN $\mathsf{v}_j$ and CN $\mathsf{c}_i$ is then drawn if client $j$ participates in test group $i$, i.e., if $a_{i,j} = 1$. Matrix $\boldsymbol{A}$—and hence the corresponding bipartite graph—is a design choice that may be decided offline, analogous to the model architecture, and shared with the clients for transparency.

*Example* 1. The bipartite graph corresponding to a scenario with 5 clients and 2 test groups with assignment matrix $\boldsymbol{A} = \begin{pmatrix} 1 & 1 & 0 & 1 & 0 \\ 0 & 1 & 1 & 0 & 1 \end{pmatrix}$ is depicted in Fig. 2a.

In FedGT, for each test group, a secure aggregation mechanism is employed to reveal only the aggregate of the client models in the test group to the server. Let $\boldsymbol{u}_i$, $i \in [m]$, be the aggregate model of test group $i$. The central server then applies a test $\mathsf{t} : \boldsymbol{u} \to \{0, 1\}$ to the aggregate model for each test group to be used for the identification of malicious clients in the group. Let $t_i = \mathsf{t}(\boldsymbol{u}_i) \in \{0, 1\}$ be the result of the test for test group $i$, where $t_i = 1$ if the test is positive and $t_i = 0$ if the test is negative. We collect the result of the $m$ tests into the binary vector $\boldsymbol{t} = (t_1, t_2, \ldots, t_m)$. We remark that the proposed framework is general and can be applied to any test on the test group aggregates.

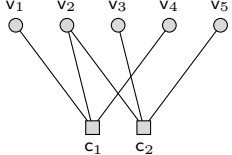
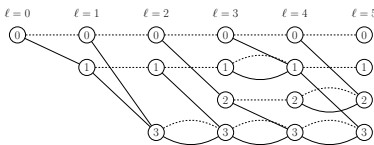

(a) Circles represent VNs, squares CNs    (b) Dashed edges represent a "0", solid edges a "1"

Figure 2: Bipartite graph (left) and trellis (right) representation of the assignment matrix $\boldsymbol{A}$ in Example 1.

We define the syndrome vector $\boldsymbol{s} = (s_1, \ldots, s_m)$, where $s_i = 1$ if at least one client participating in test group $i$ is malicious and $s_i = 0$ if no client participating in test group $i$ is malicious, i.e.,

$$s_i = \bigvee_{j \in \mathcal{P}_i} d_j \quad \text{and} \quad \boldsymbol{s} = \boldsymbol{d} \vee \boldsymbol{A}^\mathsf{T},$$

where $\vee$ is the logical disjunction.

For perfect (non-noisy) test results, it follows that $\boldsymbol{t} = \boldsymbol{s}$. However, note that the result of a test may be erroneous, i.e., the result of the test may be $t_i = 1$ even if no malicious clients are present or $t_i = 0$ even if malicious clients are present. In general, the (noisy) test vector $\boldsymbol{t}$ is statistically dependent on the syndrome vector $\boldsymbol{s}$ according to an (unknown) probability distribution $Q(\boldsymbol{t}|\boldsymbol{s})$.

Given the test results $\boldsymbol{t}$ and the assignment matrix $\boldsymbol{A}$, the goal of FedGT is to identify the malicious clients, i.e., infer the defective vector $\boldsymbol{d}$. The design of the assignment matrix $\boldsymbol{A}$ and the corresponding inference problem is akin to an error-correcting coding problem, where the assignment matrix $\boldsymbol{A}$ can seen as the parity-check matrix of a code, and the inference problem corresponds to a decoding operation based on $\boldsymbol{A}$ and $\boldsymbol{t}$. Thus, a suitable choice for $\boldsymbol{A}$ is the parity-check matrix of a powerful error-correcting code, i.e., with good distance properties. Furthermore, $\boldsymbol{d}$ can be inferred by applying conventional decoding techniques. We denote by $\hat{\boldsymbol{d}} = (\hat{d}_1, \ldots, \hat{d}_n)$ the estimated defective vector provided by the decoding operation, and define $\hat{\mathcal{M}} = \{i : \hat{d}_i = 1\}$. Once $\hat{\boldsymbol{d}}$ has been obtained, clients $i \in \hat{\mathcal{M}}$ are excluded from the training and the server aggregates the models of the remaining—flagged non-malicious—clients by means of secure aggregation.

The performance of FedGT, measured in terms of the utility of the model, is affected by two quantities: the misdetection probability, i.e., the probability that a malicious client is flagged as non-malicious, and the false-alarm probability, i.e., the probability that a non-malicious client is flagged as malicious, defined as

$$P_{\mathsf{MD}} \triangleq \frac{1}{n} \sum_{i=1}^{n} \Pr(\hat{d}_i = 0 | d_i = 1), \qquad P_{\mathsf{FA}} \triangleq \frac{1}{n} \sum_{i=1}^{n} \Pr(\hat{d}_i = 1 | d_i = 0).$$

A high misdetection probability will result in many malicious clients poisoning the global model, hence yielding poor utility, while a high false-alarm probability will result in excluding many non-malicious clients from the training, thereby also impairing the utility. The misdetection and false-alarm probabilities depend in turn on the assignment matrix $\boldsymbol{A}$, the decoding strategy, and the nature of the test performed. We discuss the decoding strategy to estimate $\boldsymbol{d}$ in Section 5.

### 4.1 Privacy-security trade-off and the choice of assignment matrix $\boldsymbol{A}$

Vanilla FL (McMahan et al., 2017) and FL with *full* secure aggregation (Bonawitz et al., 2017) can be seen as the two extreme cases of FedGT, corresponding to $n$ (non-overlapping) groups and a single group with $n$ clients, respectively. In vanilla FL, tests on individual models can be conducted, facilitating the identification of malicious clients. However, this comes at the expense of clients' privacy. In contrast, full secure aggregation provides privacy by enabling the server to observe only the aggregation of the $n$ models, but it does not permit the identification of malicious clients.

Under FedGT, the server observes $m$ aggregated models $\boldsymbol{u}_1 \ldots, \boldsymbol{u}_m$, with $\boldsymbol{u}_i = \sum_{j=1}^{n} a_{i,j} \boldsymbol{c}_j$ and $\boldsymbol{c}_j$ being the local model of client $j$. The privacy of the clients increases with the number of models aggregated (Elkordy et al., 2022). Hence, FedGT trades privacy for providing security, i.e., identification of the malicious clients. Furthermore, due to the aggregates being from overlapping

groups, there might be additional privacy loss. This loss depends on the assignment matrix $\boldsymbol{A}$ and is agnostic to the number of malicious clients participating in the training. The privacy of FedGT is given in the following proposition (the proof is given in Appendix E).

**Proposition 1.** *Let the assignment of clients to test groups be defined by assignment matrix $\boldsymbol{A}$ and let $r$ be the smallest non-zero Hamming weight of the vectors in the row span of $\boldsymbol{A}$ (in the coding theory jargon, the minimum Hamming distance of the code generated by $\boldsymbol{A}$ as its generator matrix). Then FedGT achieves the same privacy as a secure aggregation scheme with $r$ clients.*

The assignment matrix $\boldsymbol{A}$ should be carefully chosen to optimize the trade-off between privacy and security: To improve identification of malicious clients, one should choose $\boldsymbol{A}$ as the parity-check matrix of an error-correcting code with good distance properties, while to achieve a high privacy level, $\boldsymbol{A}$ should correspond to the generator matrix of a code of large minimum Hamming distance. Such codes are readily available, see, MacWilliams & Sloane (1977).

## 5    DECODING: INFERRING THE DEFECTIVE VECTOR $\boldsymbol{d}$

Several decoding techniques can be applied to infer $\boldsymbol{d}$. In this paper, we consider optimal inference in a Neyman-Pearson sense (see Appendix C), which prescribes for some $\Delta' > 1$

$$\hat{d}_i = \left\{ \begin{array}{ll} 0 & \text{if} \ \ \Pr(\boldsymbol{t}|d_i = 0) > \Pr(\boldsymbol{t}|d_i = 1)\Delta' \\ 1 & \text{if} \ \ \Pr(\boldsymbol{t}|d_i = 0) < \Pr(\boldsymbol{t}|d_i = 1)\Delta' \end{array} \right. .$$

The Neyman-Pearson criterion can be rewritten in terms of the log-likelihood ratio (LLR) $L_i = \log(\Pr(\boldsymbol{t}|d_i = 0)/\Pr(\boldsymbol{t}|d_i = 1))$ as

$$\hat{d}_i = \left\{ \begin{array}{ll} 0 & \text{if} \ \ L_i > \Delta \\ 1 & \text{if} \ \ L_i < \Delta \end{array} \right. , \tag{1}$$

where $\Delta = \log(\Delta')$. Further, we can write the LLR $L_i$ as

$$L_i = \log\left( \frac{\Pr(d_i = 0|\boldsymbol{t})}{\Pr(d_i = 1|\boldsymbol{t})} \right) - \log\left( \frac{\Pr(d_i = 0)}{\Pr(d_i = 1)} \right) = L_i^{\mathsf{APP}} - \log\left( \frac{1-\delta}{\delta} \right) , \tag{2}$$

where $\delta$ is the *prevalence* of malicious clients in the population of $n$ clients, i.e., the probability of a client being malicious, $\delta = \Pr(d_i = 1)$. In a frequentist approach to probability, $\delta = n_{\mathsf{m}}/n$. Using (2), the Neyman-Pearson criterion in (1) can be rewritten in terms of the a posteriori LLR $L_i^{\mathsf{APP}}$ as

$$\hat{d}_i = \left\{ \begin{array}{ll} 0 & \text{if} \ \ L_i^{\mathsf{APP}} > \Lambda \\ 1 & \text{if} \ \ L_i^{\mathsf{APP}} < \Lambda \end{array} \right. , \tag{3}$$

where $\Lambda = \Delta + \log\left(\frac{1-\delta}{\delta}\right)$. Note that, as $\Lambda$ increases, $P_{\mathsf{FA}}$ increases and $P_{\mathsf{MD}}$ decreases.

The Neyman-Pearson criterion requires the computation of the a posteriori LLRs $L_i^{\mathsf{APP}}$. For not-too-large matrices $\boldsymbol{A}$, they can be computed efficiently via the forward-backward algorithm (Bahl et al., 1974), which exploits the trellis representation of the assignment matrix $\boldsymbol{A}$. For large matrices $\boldsymbol{A}$, the computation of the a posteriori LLRs is not feasible, and one needs to resort to suboptimal decoding strategies, such as belief propagation (Kschischang et al., 2001) (see Appendix L, Item 3).

In Subsection 5.1, we describe how to obtain the trellis diagram corresponding to a given assignment matrix $\boldsymbol{A}$, and in Subsection 5.2, we discuss the forward-backward algorithm to compute the a posteriori LLRs to infer $\boldsymbol{d}$.

### 5.1    TRELLIS REPRESENTATION OF ASSIGNMENT MATRIX $\boldsymbol{A}$

In this section, we describe the trellis representation corresponding to assignment matrix $\boldsymbol{A}$, which can be used to compute the a posteriori LLRs as described in Section 5.2. The trellis representation was originally introduced for linear block codes in Wolf (1978) and applied to group testing in Liva et al. (2021).

For a given defective vector $\tilde{\boldsymbol{d}}$ (not necessarily the true one), define the *syndrome vector* $\tilde{\boldsymbol{s}} = (\tilde{s}_1, \ldots, \tilde{s}_n)$, where $\tilde{s}_i$ is given by $\tilde{s}_i = \bigvee_{j \in \mathcal{P}_i} \tilde{d}_j$. The syndrome vector can be written as a function

of the defective vector $\tilde{\boldsymbol{d}}$ and the assignment matrix as $\tilde{\boldsymbol{s}} = \tilde{\boldsymbol{d}} \vee \boldsymbol{A}^{\mathsf{T}}$. Note that several defective vectors are compatible with a given syndrome $\tilde{\boldsymbol{s}}$. Let $\mathcal{D}$ be the set of all possible defective vectors, i.e., all binary tuples of length $n$. We denote by $\mathcal{D}_{\boldsymbol{s}}$ the set of defective vectors compatible with syndrome vector $\boldsymbol{s}$, i.e., $\mathcal{D}_{\tilde{\boldsymbol{s}}} = \{\tilde{\boldsymbol{d}} \in \mathcal{D} : \tilde{\boldsymbol{d}} \vee \boldsymbol{A}^{\mathsf{T}} = \tilde{\boldsymbol{s}}\}$.

Let $\boldsymbol{a}_j$ be the $j$-th column of matrix $\boldsymbol{A}$. The syndrome corresponding to defective vector $\tilde{\boldsymbol{d}}$ can then be rewritten as $\tilde{\boldsymbol{s}} = \bigvee_{i=1}^{n}(\tilde{d}_i \wedge \boldsymbol{a}_i^{\mathsf{T}})$, where $\wedge$ is the logical conjunction. This equation naturally leads to a trellis representation of the assignment matrix $\boldsymbol{A}$ as explained next. A trellis is a graphical way to represent matrix $\boldsymbol{A}$, consisting of a collection of nodes connected by edges. The trellis corresponding to matrix $\boldsymbol{A}$ in Example 1 is depicted in Fig. 2b. Horizontally, the nodes, called trellis states, are grouped into sets indexed by parameter $\ell \in \{0, \ldots, n\}$, referred to as the trellis depth.

Let $\tilde{\boldsymbol{s}}_\ell$ be the *partial* syndrome vector at trellis depth $\ell \in [n]$ corresponding to $\tilde{\boldsymbol{d}}$, given as $\tilde{\boldsymbol{s}}_\ell = \bigvee_{i=1}^{\ell}(\tilde{d}_i \wedge \boldsymbol{a}_i^{\mathsf{T}})$. It is easy to see that $\tilde{\boldsymbol{s}}_\ell$ can be obtained from $\tilde{\boldsymbol{s}}_{\ell-1}$ as $\tilde{\boldsymbol{s}}_\ell = \tilde{\boldsymbol{s}}_{\ell-1} \vee (\tilde{d}_\ell \wedge \boldsymbol{a}_\ell^{\mathsf{T}})$, with $\tilde{\boldsymbol{s}}_0$ being the all-zero vector. The trellis representation is such that each state in the trellis represents a particular partial syndrome. The trellis is then constructed as follows: At trellis depth $\ell = 0$ there is a single trellis state corresponding to $\tilde{\boldsymbol{s}}_0$. At trellis depth $\ell \in [n]$, the trellis states correspond to all possible partial syndrome vectors $\tilde{\boldsymbol{s}}_\ell$ for all possible partial syndrome vectors $(\tilde{d}_1, \ldots, \tilde{d}_\ell)$, with $\tilde{d}_i \in \{0, 1\}$. For example, at trellis depth $\ell = 1$ there are only two trellis states, corresponding to partial syndromes $0 \wedge \boldsymbol{a}_1^{\mathsf{T}} = (0, \ldots, 0)$ and $1 \wedge \boldsymbol{a}_1^{\mathsf{T}} = (a_{1,1}, \ldots, a_{1,m})$, i.e., for $\tilde{d}_1 = 0$ and $\tilde{d}_1 = 1$, respectively. Note that at trellis depth $\ell = n$, there are $2^m$ trellis states, corresponding to all possible syndromes $\tilde{\boldsymbol{s}}$. For simplicity, we label the trellis state corresponding to partial syndrome vector $\tilde{\boldsymbol{s}}_\ell = (s_{\ell,1}, \ldots, s_{\ell,m})$ by its decimal representation $\sum_{i=1}^{m} \tilde{s}_{\ell,i} 2^{i-1}$. Finally, an edge from the node at trellis depth $\ell$ corresponding to partial syndrome $\tilde{\boldsymbol{s}}_\ell$ to the node at trellis depth $\ell+1$ corresponding to partial syndrome $\tilde{\boldsymbol{s}}_{\ell+1}$ is drawn if $\tilde{\boldsymbol{s}}_{\ell+1} = \tilde{\boldsymbol{s}}_\ell \vee (\tilde{d}_{\ell+1} \wedge \boldsymbol{a}_{\ell+1}^{\mathsf{T}})$, with $\tilde{d}_{\ell+1} \in \{0, 1\}$. The edge is labeled by the value of $\tilde{d}_{\ell+1}$ enabling the transition between $\tilde{\boldsymbol{s}}_\ell$ and $\tilde{\boldsymbol{s}}_{\ell+1}$.

*Example* 2. For the trellis of Fig. 2b, corresponding to the assignment matrix $\boldsymbol{A}$ in Example 1 with $n = 5$ nodes and $m = 2$ tests, the number of trellis states at trellis depth $\ell = 5$ is $2^2 = 4$, i.e., all length-2 binary vectors (in decimal notation $\{0, 1, 2, 3\}$). At trellis depth $\ell = 2$, there are three states, corresponding to all possible partial syndromes $\tilde{\boldsymbol{s}} = \bigvee_{i=1}^{2}(\tilde{d}_i \wedge \boldsymbol{a}_i^{\mathsf{T}})$, i.e., all possible (binary) linear combinations of the two first columns of matrix $\boldsymbol{A}$, resulting in states $(0, 0) \vee (0, 0) = (0, 0) = 0$, $(0, 0) \vee (1, 1) = (1, 1) = 3$, $(1, 0) \vee (0, 0) = (1, 0) = 1$, and $(1, 0) \vee (1, 1) = (1, 1) = 3$.

The trellis graphically represents all possible defective vectors $\tilde{\boldsymbol{d}}$ and their connection to the syndromes $\tilde{\boldsymbol{s}}$ via the assignment matrix $\boldsymbol{A}$. In particular, the paths along the trellis originating in the all-zero state at trellis depth $\ell = 0$ and ending in trellis state $\tilde{\boldsymbol{s}}$ at trellis depth $\ell = n$ correspond to all defective vectors $\tilde{\boldsymbol{d}}$ compatible with syndrome $\tilde{\boldsymbol{s}}$.

## 5.2 THE FORWARD-BACKWARD ALGORITHM

The a posteriori LLRs can be computed efficiently using the trellis representation of matrix $\boldsymbol{A}$ introduced in the previous subsection via the forward-backward algorithm (Bahl et al., 1974). Let $\mathcal{E}_\ell^{(0)}$ and $\mathcal{E}_\ell^{(1)}$ be the set of edges connecting trellis states at trellis depth $\ell - 1$ with states at trellis depth $\ell$ labeled by $\tilde{d}_\ell = 0$ and $\tilde{d}_\ell = 1$, respectively. $L_\ell^{\mathsf{APP}}$ can be computed as

$$L_\ell^{\mathsf{APP}} = \log \sum_{(\sigma', \sigma) \in \mathcal{E}_\ell^{(0)}} \alpha_{\ell-1}(\sigma')\gamma(\sigma', \sigma)\beta_\ell(\sigma) - \log \sum_{(\sigma', \sigma) \in \mathcal{E}_\ell^{(1)}} \alpha_{\ell-1}(\sigma')\gamma(\sigma', \sigma)\beta_\ell(\sigma), \quad (4)$$

where $(\sigma', \sigma)$ denotes an edge connecting state $\sigma'$ at trellis depth $\ell - 1$ with state $\sigma$ at trellis depth $\ell$.

The quantities $\alpha_{\ell-1}(\sigma')$ and $\beta_\ell(\sigma)$ are called the forward and backward metrics, respectively, and can be computed using the recursions

$$\alpha_\ell(\sigma) = \sum_{\sigma'} \alpha_{\ell-1}(\sigma')\gamma_\ell(\sigma', \sigma), \qquad \beta_{\ell-1}(\sigma') = \sum_{\sigma} \beta_\ell(\sigma)\gamma_\ell(\sigma', \sigma),$$

with initialization of the forward recursion $\alpha_0(0) = 1$ and of the backward recursion $\beta_n(\sigma) = Q(\boldsymbol{t}|\boldsymbol{s}(\sigma))$, where $\boldsymbol{s}(\sigma)$ is the syndrome corresponding to trellis state $\sigma$. The quantity $\gamma_\ell(\sigma', \sigma)$ is

called the branch metric and is given by

$$\gamma_\ell(\sigma', \sigma) = \begin{cases} 1 - \delta & \text{if } (\sigma', \sigma) \in \mathcal{E}_\ell^{(0)} \\ \delta & \text{if } (\sigma', \sigma) \in \mathcal{E}_\ell^{(1)} \end{cases}.$$

The a posteriori LLRs computed via (4) are then used to make decisions on $\{d_i\}$ according to (3).

## 6 EXPERIMENTS

**Experiment configuration and hyperparameters.** We consider a cross-silo scenario with $n = 15$ clients (all participating in each training round) out of which $n_{\mathrm{m}}$ are malicious. In Appendix J, we also provide results for $n = 31$ clients. The goal of the server is to prevent an attack by identifying the malicious clients and exclude their models from the global aggregation. The experiments are conducted for image classification problems on the MNIST (LeCun & Cortes, 2010), CIFAR-10 (He et al., 2016), and ISIC2019 (Codella et al., 2018) datasets for which we rely on a single-layer neural network, a ResNet-18 (Krizhevsky et al., 2009), and an Efficientnet-B0 pretrained on Imagenet (Tan & Le, 2019), respectively.

Similar to previous works (Pan et al., 2020; Cao et al., 2020; Mallah et al., 2021; Park et al., 2021; Nguyen et al., 2022), we assume that the server has a small validation dataset at its disposal to perform the group tests (the validation dataset is not used for training). Such dataset is not required by FedGT, but is used here due to our choice for the tests in the experiments. The validation dataset should contain data that are sampled from a distribution close to the underlying distribution of the (benign) clients' datasets, i.e., it should be a *quasi-dataset* (Pan et al., 2020; Mallah et al., 2021). For the experiments, we create the validation dataset by randomly sampling 100 data-points from the training set, i.e., the label distribution may not be uniform. For MNIST and CIFAR10, the remaining training data (of size 59900 and 49900) are split evenly at random among the 15 clients, resulting in homogenous data among the clients. For ISIC2019, we follow (Ogier du Terrail et al., 2022) and randomly partition the dataset into a training and a test set consisting of 19859 and 3388 samples, respectively. We then partition the training dataset into six parts according to the image acquisition system used to collect the images. Finally, we iteratively split the largest partition in half until we have 15 partitions. As shown in Appendix K, this procedure results in a heterogeneous setting where both label distributions and number of samples differ among clients.

For MNIST, we use the cross-entropy loss and stochastic gradient descent with a learning rate of 0.01, batch size of 64, and number of local epochs equal to 1. For CIFAR-10, we use the cross-entropy loss and stochastic gradient descent with momentum and parameters taken from McMahan et al. (2017): the learning rate is 0.05, momentum is 0.9, and the weight decay is 0.001. Furthermore, the batch size is set to 128 and the number of local epochs is set to 5. The results presented are averaged over 100 and 5 runs for MNIST and CIFAR-10, respectively, and we discuss the statistical variations of the results in Appendix F. For ISIC2019, we use the focal loss in (Lin et al., 2017) and stochastic gradient descent with a learning rate of 0.0005, momentum of 0.9, and weight decay equal to 0.0001. The batch size equals 64 and the number of local epochs is set to 1. Furthermore, we use the same set of augmentations as in (Ogier du Terrail et al., 2022) to encourage generalization during the training.

We compare the performance of FedGT with three benchmarks: "no defense", "oracle" and RFA (Pillutla et al., 2022). The no defense benchmark corresponds to the case of plain FL over all clients, i.e., disregarding some clients may be malicious, while the oracle assumes that the server knows the malicious clients and discards them. Note that RFA belongs to a short list of defense mechanisms that also provide privacy.

To demonstrate the effectiveness of FedGT, we perform the group testing step only once during the training. Note that this constitutes the weakest version of our framework as the group testing may be performed in each round at the expense of increased communications. In particular, for MNIST, we perform a group test in the first round and for CIFAR-10 and ISIC2019, in the fifth round. We pick as the assignment matrix a parity-check matrix of a BCH code (Bose & Ray-Chaudhuri, 1960) of length 15 and redundancy 8, meaning that we create a group testing scheme where the 15 clients are pooled into 8 groups, each containing 4 clients.

**Privacy.** With our choice of assignment matrix, FedGT guarantees the same level of privacy of full secure aggregation with 4 clients. This stems from the property that any linear combination of

the server's group aggregates leads to an aggregation involving no fewer than $4$ client models, as elucidated in Proposition 1. We provide the learning accuracy associated with alternative assignment matrix choices, each yielding distinct privacy guarantees, in Appendix I.

**Test design.** Let $v_i$ denote a measured metric on the aggregated model of test group $i$ evaluated on the validation dataset and let $\boldsymbol{v} = (v_1, v_2, \dots, v_m)$. To determine whether a client is malicious or not, we consider a simple approach where

$$t_i = \begin{cases} 0 & \text{if } v_i \geq \rho \cdot v^\star \\ 1 & \text{otherwise} \end{cases}, \quad i \in [m], \tag{5}$$

where $0 \leq \rho \leq 1$ and $v^\star = \arg\max_{i \in [m]} v_i$, i.e., a test group is labeled as negative when the metric is close to the maximum value among the test groups. Thereafter, the test vector $\boldsymbol{t} = \{t_1, t_2, \dots t_m\}$ is fed to a forward-backward decoder as described in Section 5.2 and $\hat{\boldsymbol{d}}$ is obtained. In the special case of $\hat{\boldsymbol{d}} = \mathbf{1}$, i.e., all clients are flagged as malicious, one may either cancel the learning procedure or, as done in the experiments, proceed without a defense. For all experiments, we use $\rho = 0.96$. We remark that FedGT is compatible with any testing strategy.

**Distribution $Q(\boldsymbol{t}|\boldsymbol{s})$, prevalence $\delta$, and threshold $\Lambda$.** The decoder requires the distribution $Q(\boldsymbol{t}|\boldsymbol{s})$, the prevalence $\delta$, and the threshold $\Lambda$. We model the noisiness of the test as $Q(\boldsymbol{t}|\boldsymbol{s}) = \prod_{i=1}^{m} Q(t_i|s_i)$, and $Q(t_i|s_i)$ as a binary symmetric channel (BSC), i.e., $Q(t_i|s_i) = 1 - p$ if $t_i = s_i$ and $Q(t_i|s_i) = p$ if $t_i \neq s_i$. Note that the true distribution $Q(\boldsymbol{t}|\boldsymbol{s})$ is in general unknown and hard to estimate—even assuming that the BSC is a good model, $p$ is unknown and test dependent. In our experiments, we arbitrarily choose $p = 0.05$. However, in Appendix G, we demonstrate that FedGT is robust to different choices of $p$. Moreover, in the experiments we feed the decoder the prevalence, i.e., prior probability, $\delta = n_{\mathsf{m}}/n$. For the case of $n_{\mathsf{m}} = 0$, we use a mismatched metric $\delta = 0.1$. Similarly to $p$, $n_{\mathsf{m}}$ is unknown and we demonstrate in Appendix G that FedGT is robust to a mismatch in the prevalence fed to the decoder. Finally, for all three datasets we evaluate FedGT for the threshold $\Lambda \in \{0.1, 0.2, \dots, 0.9\}$. Although the best performance of FedGT is obtained by optimizing $\Lambda$ for the specific context, we report our results for the same $\Lambda$ over all values of $n_{\mathsf{m}}$. We reiterate that $\Lambda$ is a design choice and trades off between misdetections and false alarms. We provide results for optimized values of $\Lambda$ in Appendix A.1.

### 6.1 EXPERIMENTAL RESULTS FOR TARGETED ATTACKS

For targeted data-poisoning, we consider label-flipping attacks. We refer to the attacked label as the source label and the resulting label after the flip as the target label. For MNIST, we consider malicious clients to flip source label 1 into target label 7. As such, the objective of the malicious clients is to cause the global model to misclassify 1's into 7's. Similarly, for CIFAR-10, malicious clients change source label 7, i.e., horses, into target label 4, i.e., deers. For the ISIC2019 dataset, malicious clients mislabel source label 0, i.e., melanoma, into target label 1, i.e., mole. Note that this attack has a huge medical impact, as the goal of the attacker is to force the model to classify cancer into non-cancer. As the adversary objective is not to deteriorate the global model but only to make it misinterpret the source label as the target label, we adopt the *attack accuracy* as the metric of interest, defined as the fraction of source labels that are classified as the target label in the test dataset.

We consider the source label recall, i.e., the fraction of source labels that are classified into the correct label, as the performance metric adopted in the testing strategy (5) to flag test groups containing malicious clients. Although we assume the server is informed of what label is under attack, we remark that it is straightforward to detect the labels under attack as the source label recall in a test group is significantly reduced in the presence of a malicious client.

In Fig. 1, we show the attack accuracy (evaluated over the test dataset) under FedGT. For MNIST (Fig. 1a), we observe a modest impact of the label flip, even for $n_{\mathsf{m}} \in [5]$. Nevertheless, FedGT significantly diminishes the attack accuracy compared to no defense and performs close to oracle. RFA slightly outperforms FedGT. However, we recall that it lacks the capability of identifying malicious clients and it has a higher communication cost (discussed in Appendix B). For CIFAR10, the label flip attack has a significant impact as can be seen from the no-defense attack accuracy being close to $40\%$ when $1/3$ of the clients are malicious. FedGT performs significantly better than RFA for $n_{\mathsf{m}} \geq 3$ and close to the oracle—FedGT reduces the attack accuracy to $7\%$ when $1/3$ of the clients are malicious (compared to $19.34\%$ for RFA).

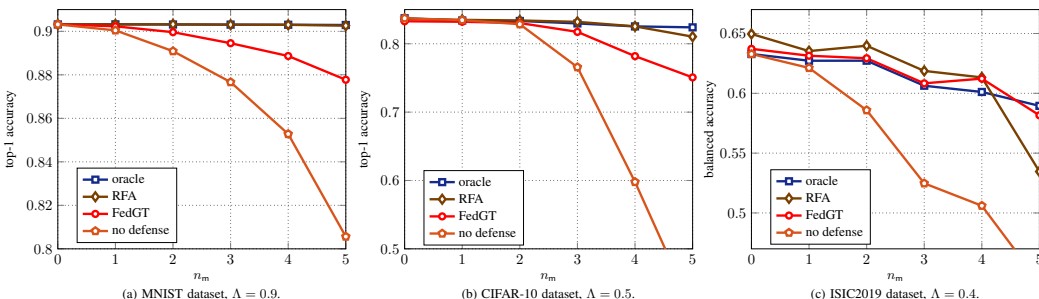

Figure 3: Average top-1 accuracy versus the number of malicious clients

For ISIC2019, RFA achieves only a small improvement in attack accuracy with respect to no defense and performs as no defense for $n_{\mathrm{m}} = 5$ (RFA is known to perform poorly for heterogeneous data across clients (Li et al., 2023)). FedGT significantly diminishes the attack accuracy and performs even better than oracle. This can be explained from the data heterogeneity across clients where some clients, although not malicious, will be biased to output a given label, see, e.g., client 4 and client 10 in Appendix K. Hence, due to the testing strategy, FedGT may identify benign clients exhibiting extreme heterogeneity as malicious to be removed from the training, ultimately benefiting the overall utility of the global model. In Appendix A, we show how the attack accuracy evolves with the number of communication rounds and we provide the top-1 (or balanced) accuracy of the global model.

### 6.2 EXPERIMENTAL RESULTS FOR UNTARGETED ATTACKS

We consider a label permutation attack where malicious clients offset their data labels by 1, i.e., $L_{\mathrm{new}} = (L_{\mathrm{old}} + 1) \mod n_{\mathrm{c}}$, where $n_{\mathrm{c}}$ is the number of classes. The attack aims at damaging the classification accuracy as a whole, i.e., an attacker wants to lower the top-1 accuracy (MNIST and CIFAR-10) or the balanced accuracy (ISIC2019). For this reason, we use the top-1 accuracy (MNIST and CIFAR10) and the balanced accuracy (ISIC2019) on the test group's aggregates as the metric for the testing strategy in (5) adopted by the server. The balanced accuracy is the average recall per class, used to take into account class imbalances, as in the case of ISIC2019 (Ogier du Terrail et al., 2022).

In Fig. 3, we plot the top-1 accuracy versus $n_{\mathrm{m}}$ for MNIST (left) and CIFAR-10 (middle), and the balanced accuracy for ISIC2019 (right). For all cases, when no defense is in place, a significant drop in accuracy is observed as the number of malicious clients grows. FedGT permits to identify the malicious clients with low misdetection and false alarm probabilities, which translates into a high accuracy: For $n_{\mathrm{m}}$ up to 3, the top-1 accuracy of FedGT is at most 1% lower than the performance of an oracle for CIFAR-10 and MNIST. Furthermore, in Fig. 3c we observe that FedGT performs close to the oracle for all values of $n_{\mathrm{m}}$ for experiments over ISIC2019. The robust performance of RFA is anticipated due to our untargeted attack rendering malicious client models significantly different from benign models. Consequently, the geometric median—essentially performing a majority vote—assigns the malicious models a very low weight. However, even with the simple test in (5) only being performed in a single round, FedGT achieves a performance close to that of RFA and it even outperforms RFA for $n_{\mathrm{m}} = 5$ for the ISIC2019 dataset.

## 7 CONCLUSION

We proposed FedGT, a novel framework for identifying malicious clients in FL that is compatible with secure aggregation. By grouping clients into overlapping groups, FedGT enables the identification of malicious clients at the expense of secure aggregation involving fewer clients. Experiments on a cross-silo scenario for different data-poisoning attacks demonstrate the effectiveness of FedGT in identifying malicious clients, resulting in high model utility and low attack accuracy. Remarkably, FedGT significantly outperforms RFA in several scenarios. To the best of our knowledge, this is the first work that provides a solution for identifying malicious clients in FL with secure aggregation.

In this paper, we considered a cross-silo scenario, wherein the number of clients is moderate. The extension of FedGT to a cross-device scenario, where the number of clients is typically very large, is discussed in Appendix L. Moreover, our framework is also applicable against model-poisoning attacks provided proper group tests are available (see Appendix L).

## ETHICS STATEMENT

In this work, our studies and experiments are not related to human subjects, practices to data set releases, discrimination/bias/fairness concerns, and also do not have legal compliance or research integrity issues. The datasets we use are public and the attacks performed are done only for experimental purposes. Poisoning attacks are attacks performed by malicious users that aim to deteriorate the global model's performance in the federation. In our work, we aim at privately identifying the malicious clients and excluding them from the learning, such that the effects of the poisoning attacks can be hindered.

## REPRODUCIBILITY STATEMENT

We provide extensive details regarding the parameters and strategies we use to produce our results. The source code we used to generate the results is available via the anonymous link `https://anonymous.4open.science/r/4D37/`.

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

APPENDIX

We discuss some important properties of our proposed framework, FedGT, that we could not include in the main body of the paper due to space limitations. The appendix is organized as follows:

- In Appendix A, we present more results obtained by the simulations as described in Section 6.

- Appendix B discusses the communication cost of FedGT

- In Appendix C we state the Neyman-Pearson decision criterion.

- In Appendix D, we give the trellises for vanilla federated learning and federated learning with full secure aggregation. We believe this section will help the reader better understand the trellis decoding and the trade-off between privacy and security of our framework.

- We prove Proposition 1 in Appendix E

- Appendix F discusses the statistical variations of the results presented in Fig. 1 and Fig. 3 of the main paper.

- Throughout the paper, we have presented results obtained by feeding the decoder a frequentist choice of prevalence, $\delta = n_{\mathsf{m}}/n$. Hence, the decoder, i.e., the server, knows the number of malicious clients $n_{\mathsf{m}}$—an unrealistic assumption in practice. Moreover, the decoder models the noisiness of the tests with a binary symmetric channel with crossover probability $p$ whose real value is unknown (even in our experiments). In Appendix G, we show the robustness of FedGT to different values of $p$ and a mismatch in prevalence, denoted by $\tilde{\delta}$.

- In Appendix H, we discuss potential testing strategies to improve the performance of FedGT and present results of FedGT with a perfect testing strategy, which we refer to as *noiseless* FedGT.

- In Appendix I, we present numerical results that capture the privacy-security trade-off.

- In Appendix J, we show numerical results for a larger scenario including 31 clients.

- In Appendix K, we provide more details around the experiments pertaining to the ISIC2019 dataset.

- Appendix L underscores the constraints inherent to our framework and puts forth suggestions for mitigating these limitations.

## A  FURTHER EXPERIMENTAL RESULTS

In this section we present more results obtained from experiments as detailed in Section 6 for both targeted and untargeted attacks, and for all three datasets (MNIST, CIFAR-10, ISIC2019). We show how the learning accuracy (balanced, top-1, or attack accuracy) evolves with respect to communication rounds. For ease of presentation, in this section we focus on the case $n_{\mathsf{m}} \in \{1, 3, 5\}$.

**Targeted attacks.** In Fig. 4, we plot the attack accuracy of a label-flip attack over communication rounds. For MNIST, the attack impact is limited even for the case of no defense. However, even though the attack is weak, FedGT is able to detect the malicious clients and dampen the attack, e.g., after 10 communication rounds for $n_{\mathsf{m}} = 5$, the attack accuracy is more than three times weaker by employing FedGT compared to no defense. The robust aggregation strategy RFA in (Pillutla et al., 2022) achieves better attack accuracy than FedGT (we recall that, contrary to FedGT, RFA lacks the capability to identify malicious clients), but our proposed framework is very competitive in the $n_{\mathsf{m}} \leq 3$ regime.

For CIFAR-10, the impact of the attack is more significant. For example, for $n_{\mathsf{m}} = 5$, the oracle-aided scheme has an attack accuracy of $3\%$ after 50 communication rounds, while the no defense strategy suffers an attack accuracy of $30 - 40\%$. RFA reduces the attack accuracy to $20\%$, still operating closer to the no defense strategy. On the other hand, FedGT is able to reduce the attack accuracy to $7 - 8\%$, operating much closer to the oracle-aided strategy and significantly outperforming RFA (see Fig. 4f). Similarly, one can see that FedGT outperforms RFA for $n_{\mathsf{m}} = 3$ (see Fig. 4e), while for $n_{\mathsf{m}} = 1$ their performances are similar (see Fig. 4d). The label flip attack proves to be impactful for the ISIC2019 dataset—the attack accuracy of a no defense strategy doubles compared to the oracle scheme for $n_{\mathsf{m}} = 5$ (see Fig. 4i). From Figs. 4g– 4i, one can see that FedGT performs best, i.e., it

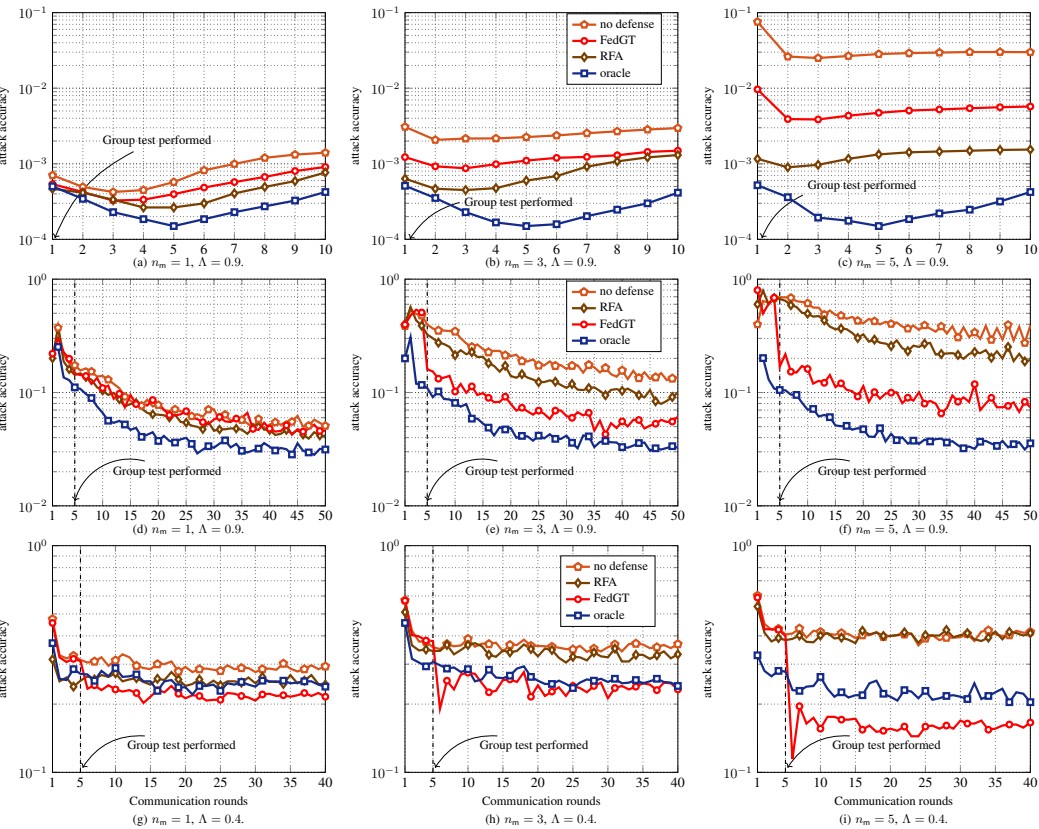

Figure 4: Average attack accuracy on the MNIST (row 1), CIFAR10 (row 2) and ISIC2019 (row 3) datasets for varying number of malicious clients.

achieves the smallest attack accuracy among the four strategies. Interestingly, FedGT outperforms oracle. This can be attributed to the heterogeneous data distribution among clients, where some clients with highly heterogeneous data points are biased and therefore deteriorate the performance of the global model.

What is often overlooked in the literature when considering targeted attacks is the impact that the defense strategy has on the utility of the global model, i.e., does the learning accuracy (top-1 or balanced) deteriorate when lower attack accuracies are achieved? Since FedGT needs to balance misdetections and false alarms, investigating the impact on the utility of the global model is important. Therefore, we provide in Table 1, 2 and 3 the values of the attack accuracy (denoted in the table as ATT) and learning accuracy (denoted as ACC; top-1 accuracy for MNIST, CIFAR-10 and balanced accuracy for ISIC2019) at the last communication round obtained from the experiments over MNIST, CIFAR-10 and ISIC2019 datasets, respectively. From Table 1, we see that RFA outperforms FedGT in terms of attack accuracy, while in terms of top-1 accuracy their differences are negligible. However, when the impact of the attack is more significant (CIFAR-10 and ISIC2019, Table 2 and 3), FedGT outperforms RFA from the attack accuracy perspective. Nevertheless, RFA achieves a higher (top-1 or balanced) accuracy, which is expected. RFA is based on the geometric median aggregation, where a majority decision strategy penalizes the outliers (minority), by assigning their model a smaller coefficient. In a targeted attack, the model weights of the malicious clients will differ only by a few entries, thus confusing the geometric median aggregation. Thus, for a relatively large number of malicious clients $n_{\mathsf{m}} \geq 3$, RFA achieves a high accuracy (top-1 or balanced)—in certain instances, RFA surpasses the oracle strategy. However, it is crucial to highlight that RFA exhibits a significantly higher attack accuracy compared to FedGT. This raises a pertinent question: should the attack accuracy or the accuracy be the figure of merit in the context of a targeted attack? To answer this question, it is important to emphasize that in our experiments with the ISIC2019 dataset, the label flip attack mislabels melanoma into benign moles, carrying devastating clinical implications. Therefore, in this case, minimizing the impact of the targeted attack should be a primary objective.

Table 1: Attack accuracy (ATT) and top-1 accuracy (ACC) measured after 10 communication rounds for experiments over MNIST dataset. The results of FedGT are obtained for $\Lambda = 0.9$.

| | Oracle | | RFA | | FedGT | | No defense | |
|---|---|---|---|---|---|---|---|---|
| $n_{\mathsf{m}}$ | ATT | ACC | ATT | ACC | ATT | ACC | ATT | ACC |
| 0 | 0.0005 | 0.903 | 0.0005 | 0.903 | 0.0005 | 0.903 | 0.0005 | 0.903 |
| 1 | 0.0004 | 0.903 | 0.0007 | 0.903 | 0.0009 | 0.903 | 0.0014 | 0.903 |
| 2 | 0.0004 | 0.902 | 0.001 | 0.903 | 0.0011 | 0.902 | 0.0016 | 0.902 |
| 3 | 0.0004 | 0.901 | 0.0013 | 0.902 | 0.0014 | 0.902 | 0.0029 | 0.901 |
| 4 | 0.0004 | 0.899 | 0.0014 | 0.902 | 0.0031 | 0.901 | 0.0093 | 0.899 |
| 5 | 0.0004 | 0.895 | 0.0015 | 0.902 | 0.0057 | 0.900 | 0.0299 | 0.895 |

Table 2: Attack accuracy (ATT) and top-1 accuracy (ACC) measured after 50 communication rounds for experiments over CIFAR-10 dataset. The results of FedGT are obtained for $\Lambda = 0.9$.

| | Oracle | | RFA | | FedGT | | No defense | |
|---|---|---|---|---|---|---|---|---|
| $n_{\mathsf{m}}$ | ATT | ACC | ATT | ACC | ATT | ACC | ATT | ACC |
| 0 | 0.0302 | 0.834 | 0.0318 | 0.837 | 0.0352 | 0.826 | 0.0302 | 0.834 |
| 1 | 0.0314 | 0.836 | 0.0414 | 0.838 | 0.046 | 0.829 | 0.0508 | 0.836 |
| 2 | 0.0318 | 0.832 | 0.057 | 0.835 | 0.0694 | 0.829 | 0.0866 | 0.832 |
| 3 | 0.0326 | 0.827 | 0.0994 | 0.828 | 0.0612 | 0.814 | 0.1348 | 0.827 |
| 4 | 0.0316 | 0.817 | 0.1404 | 0.823 | 0.0636 | 0.785 | 0.2244 | 0.817 |
| 5 | 0.0356 | 0.798 | 0.1934 | 0.819 | 0.074 | 0.775 | 0.3916 | 0.798 |

**Untargeted attacks.** In Fig. 5, we plot the top-1 accuracy (for MNIST, first row, and CIFAR-10, second row) and balanced accuracy (for ISIC2019, last row) over the communication rounds. For $n_{\mathsf{m}} = 1$, the attack is not very powerful (regardless of the dataset), and the no defense and oracle benchmarks have similar performance. For $n_{\mathsf{m}} \in \{3, 5\}$, the impact on the top-1 accuracy of the attack for MNIST and CIFAR-10 is significant, as shown by the significant gap between the no defense and oracle curves. FedGT significantly closes this gap, but RFA outperforms our strategy, due to its majority decision-based aggregation technique. We observe an interesting phenomenon for the experiments over the CIFAR-10 dataset. For $n_{\mathsf{m}} = 3$ and 5, the no defense curve exhibits significant fluctuations throughout the rounds. Furthermore, as $n_{\mathsf{m}}$ increases, the peak distance of the fluctuations also increases. This instability can be attributed to the presence of a high number of malicious clients. Notably, although the performance of FedGT also fluctuates, it does so to a significantly lesser extent. RFA also suffers from this phenomenon. For $n_{\mathsf{m}} = 5$, one can see from Fig. 5f, that the global model obtained from RFA achieves a better top-1 accuracy in round 50, but is a more unstable model (fluctuations in the top-1 accuracy) for the last 5 rounds compared to FedGT. Finally, for the ISIC2019 dataset, FedGT significantly outperforms RFA for $n_{\mathsf{m}} = 5$, while for $n_{\mathsf{m}} = 3$ RFA slightly outperforms FedGT (see Figs. 5h,5i).

## A.1 OPTIMIZATION OF THE DECODING THRESHOLD $\Lambda$

As mentioned in Section 6, we obtained the results by sweeping the decoding threshold $\Lambda \in \{0.1, 0.2, \ldots, 0.9\}$ and we present the results in Figs. 1,3-5 for a fixed threshold per dataset and attack. However, the results previously shown do not highlight the real potential of FedGT. Therefore, in Fig. 6 we plot the accuracy versus the number of malicious clients for the decoding threshold $\Lambda \in \{0.1, 0.2, \ldots, 0.9\}$ that achieves the best performance (per $n_{\mathsf{m}}$). From the experiments over

Table 3: Attack accuracy (ATT) and balanced accuracy (ACC) measured after 40 communication rounds for experiments over ISIC2019 dataset. The results of FedGT are obtained for $\Lambda = 0.4$.

| | Oracle | | RFA | | FedGT | | No defense | |
|---|---|---|---|---|---|---|---|---|
| $n_{\mathsf{m}}$ | ATT | ACC | ATT | ACC | ATT | ACC | ATT | ACC |
| 0 | 0.2587 | 0.632 | 0.2172 | 0.649 | 0.1691 | 0.642 | 0.2587 | 0.632 |
| 1 | 0.2388 | 0.632 | 0.2437 | 0.654 | 0.2155 | 0.635 | 0.2935 | 0.628 |
| 2 | 0.2437 | 0.622 | 0.2669 | 0.645 | 0.2155 | 0.632 | 0.3101 | 0.631 |
| 3 | 0.2404 | 0.614 | 0.3316 | 0.625 | 0.2321 | 0.614 | 0.3681 | 0.609 |
| 4 | 0.2106 | 0.614 | 0.3167 | 0.63 | 0.1592 | 0.613 | 0.3565 | 0.621 |
| 5 | 0.2039 | 0.615 | 0.4112 | 0.633 | 0.1658 | 0.608 | 0.4162 | 0.613 |

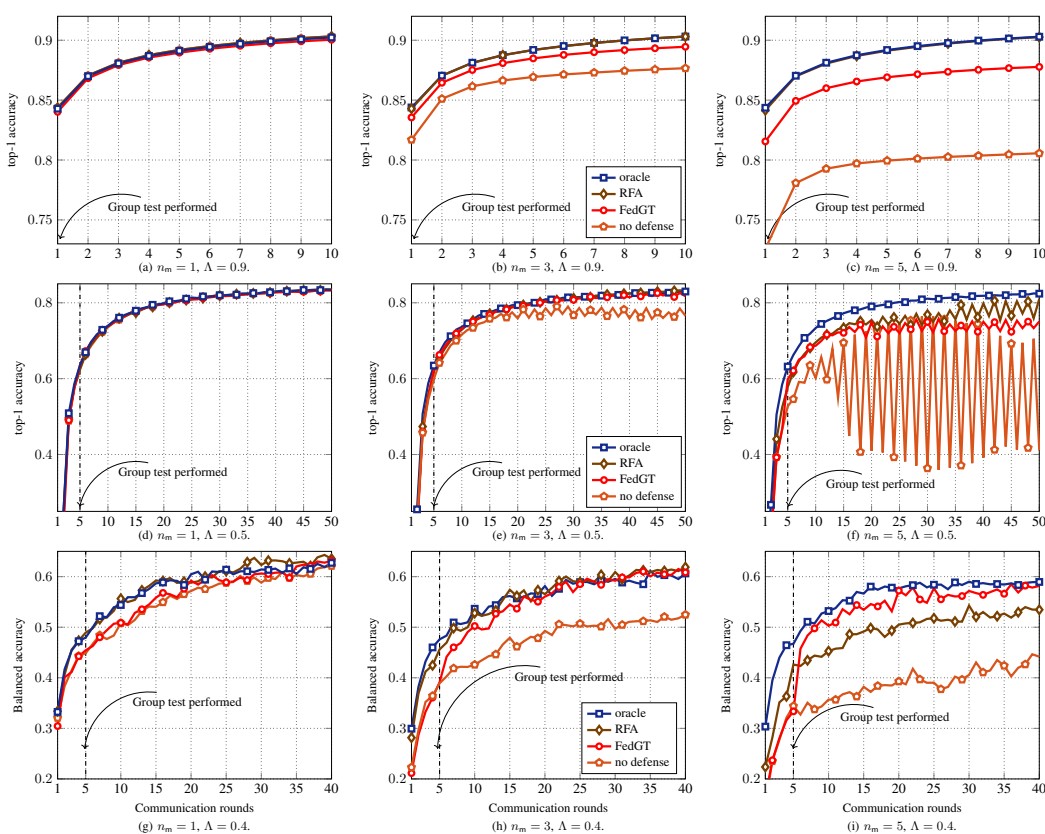

Figure 5: Average top-1 accuracy on the MNIST (row 1), CIFAR10 (row 2) and ISIC2019 (row 3) datasets for different number of malicious clients.

MNIST (Figs. 6a, 6d), we observe that the situation does not improve much compared to the fixed $\Lambda = 0.9$. However, for CIFAR-10, with the best $\Lambda$ from the set $\{0.1, 0.2, \ldots, 0.9\}$, FedGT outperforms RFA for all $n_{\mathsf{m}}$ considered (see Fig. 6b), while for untargeted attacks the top-1 accuracy is improved for $n_{\mathsf{m}} \in \{4, 5\}$, but still cannot outperform RFA (see Fig. 6e). As for experiments over ISIC2019, FedGT outperforms RFA for both attacks and for all considered values of $n_{\mathsf{m}}$ (see Figs. 6c, 6f).

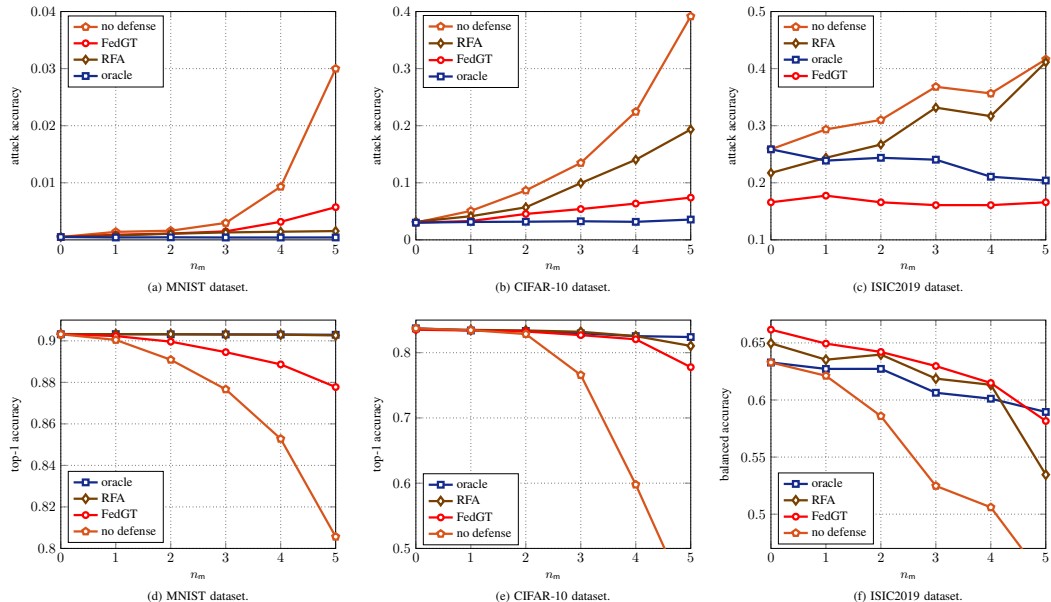

Figure 6: Performance of FedGT shown for the best decoding threshold. Its performance is compared to three strategies, no defense, oracle and RFA (robust federated aggregation). The first row represents the targeted attack, while the second one shows the performance under an untargeted attack.

## B DISCUSSION: COMMUNICATION COST

FedGT introduces a communication overhead only in the round(s) where group testing is performed. The version we have presented in this work performs group testing only in a single round, so we will focus the analysis on this particular scenario. The overall communication cost of FedGT consists of:

- **Before group testing**: Number of communication rounds $\times$ communication complexity of secure aggregation with $n$ clients.
- **Group testing round**: $m \times$ communication complexity of secure aggregation with $\max_{j \in [m]} |\mathcal{P}_j|$ clients (maximum size of groups).
- **After group testing**: Number of rounds $\times$ communication complexity of secure aggregation with $n - |\hat{\mathcal{M}}|$ clients (number of clients classified as benign from the group testing).

The communication cost in the secure aggregation of clients that are estimated benign after the group testing will vary as it depends on, e.g., the context and specific test. Therefore, here we will assume the worst-case scenario from a communication overhead perspective, which is $\hat{\mathcal{M}} = \varnothing$ and results in a communication complexity of secure aggregation of $n$ clients. The overall communication cost heavily depends on the scheme used for secure aggregation. In Jahani-Nezhad et al. (2022, Table I), the communication cost for different secure aggregation schemes is tabulated. Considering a secure aggregation scheme with linear communication complexity such as LightSecAgg (So et al., 2022), the communication cost of the group testing round is approximately $2 \times$ the complexity of secure aggregation with 15 clients for the choice of assignment matrix described in Section 6. For this particular assignment matrix, the overall communication cost in the worst-case scenario is for a single round twice the communication cost of secure aggregation with 15 clients, while all the other communication rounds have a communication cost equal to that of secure aggregation with 15 clients. Compared to RFA, which requires $3 \times$ communication cost of secure aggregation with 15 clients in each round, FedGT yields a significantly reduced communication cost.

## C THE NEYMAN-PEARSON DECISION CRITERION

The Neyman-Pearson decision criterion is a fundamental concept in statistical hypothesis testing, specifically in situations where one needs to make a decision between two competing hypotheses,

often referred to as the null hypothesis, $H_0$, and the alternative hypothesis, $H_1$. In the context of identification of malicious devices in FL, $H_0$ could correspond to 'non-malicious client' and $H_1$ to 'malicious client'. If a test chooses $H_1$ when in fact the truth was $H_0$, the error is called a false-alarm (or false positive). The error of deciding $H_0$ when $H_1$ was correct is called a misdetection (or false-negative).

The Neyman-Pearson decision criterion is optimal in the sense that it allows to minimize one type of error subject to a constraint on the other type of error. Formally, it solves the following optimization problem,

$$\max_{\text{s.t.} P_{\text{FA}} \leq \alpha} P_{\text{D}} \,, \tag{6}$$

where $P_{\text{FA}}$ is the probability of false alarm, $P_{\text{D}} = 1 - P_{\text{MD}}$ the probability of detection, and $P_{\text{MD}}$ the probability of misdetection.

This constrained optimization criterion does not require knowledge of the prior probabilities, but only a specification of the maximum allowable value for one type of error. Neyman and Pearson showed that the solution to this type of optimization is a likelihood ratio test (Neyman & Pearson, 1992).

**Lemma 1.** *Consider the likelihood ratio test (LRT)*

$$\frac{p(x|H_1)}{p(x|H_0)} \underset{H_0}{\overset{H_1}{\underset{<}{>}}} \lambda \tag{7}$$

*and define the region*

$$\mathcal{A}_{\text{LRT}}(\lambda) = \{x : p(x|H_1) > \lambda p(x|H_0)\} \,, \tag{8}$$

*i.e., the values of $x$ for which the test decides $H_1$.*

*Let the probability of false alarm for the LRT be $P_{\text{FA}}(\mathcal{A}_{\text{LRT}}(\lambda))$ and the probability of detection $P_{\text{D}}(\mathcal{A}_{\text{LRT}}(\lambda))$. In (7), $\lambda > 0$ is chosen such that $P_{\text{FA}}(\mathcal{A}_{\text{LRT}}(\lambda)) = \alpha$. Then, there does not exist another test $\mathsf{T}$ (with corresponding region $\mathcal{A}_{\mathsf{T}}$) with $P_{\text{FA}}(\mathcal{A}_{\mathsf{T}}(\lambda)) \leq \alpha$ and $P_{\text{D}}(\mathcal{A}_{\mathsf{T}}(\lambda)) > P_{\text{D}}(\mathcal{A}_{\text{LRT}}(\lambda))$, i.e., the LRT is the most powerful test with probability of false alarm less than or equal to $\alpha$.*

One may consider alternative decision criteria to the Neyman-Pearson criterion, such as Bayesian inference. In this case, the decision criterion is

$$\frac{p(H_1|x)}{p(H_0|x)} = \frac{p(H_1)}{p(H_0)} \cdot \frac{p(x|H_1)}{p(x|H_0)} \underset{H_0}{\overset{H_1}{\underset{<}{>}}} 1 \,. \tag{9}$$

In contrast to the Neyman-Pearson criterion, the Bayesian approach does not assign a special role to the null hypothesis—it treats both hypotheses equally.

In situations where striking a balance between misdetection and false alarm is important, the Neyman-Pearson criterion is generally deemed more appropriate than Bayesian inference. Specially, the Neyman-Pearson criterion is well-suited when false alarms carry significant consequences. For instance, in medical diagnoses, a false positive can result in unnecessary treatments, potentially leading to serious consequences.

As the performance of FedGT, measured in terms of the utility of the model, hinges upon a delicate balance between the probability of false alarm and the probability of misdetection, in this paper we consider optimal inference in a Neyman-Pearson sense. It is important to note that a false positive in this context involves excluding data from a benign client from the training, ultimately impairing utility. Particularly for heterogeneous data, the exclusion of benign clients can be highly detrimental. The appropriate balance between misdetection and false alarm can be achieved through careful manipulation of the parameter $\lambda$.

Furthermore, the Bayesian approach requires knowledge of the prior probabilities, which, in our scenario, are unknown.

# D  TRELLIS REPRESENTATION FOR VANILLA FEDERATED LEARNING AND FEDERATED LEARNING WITH FULL SECURE AGGREGATION

Section 5.1 describes the construction of the trellis corresponding to an assignment matrix. In Fig. 2b, we depict the trellis representation of matrix $\boldsymbol{A}$ given in Example 1, which corresponds to a group testing scheme of 5 clients and 2 groups.

In Section 4.1, we argue that there is a fundamental trade-off between privacy and security in federated learning (FL). The two extremes correspond to vanilla FL, where the server observes individual client models in each communication round, and FL with full secure aggregation. In the following, we provide the assignment matrix $\boldsymbol{A}$ and the corresponding trellis representation for both scenarios.

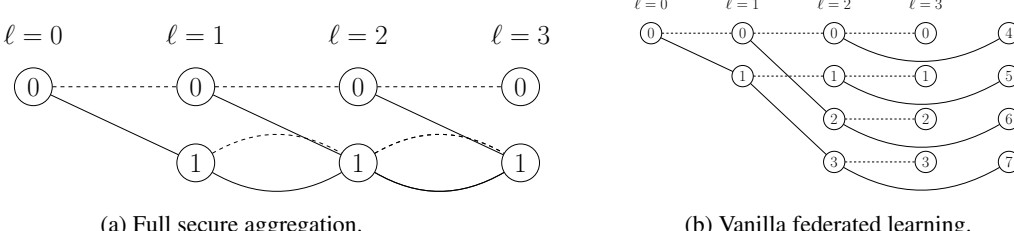

(a) Full secure aggregation.  (b) Vanilla federated learning.

Figure 7: Trellis representations of a group testing scheme with $n = 3$ nodes for the special case of *full secure aggregation* (a) and *vanilla federated learning* (b).

*Example* 3 (Full secure aggregation). In FL with full secure aggregation, the server observes only the aggregate of the models of all clients. Hence, it corresponds to the case of FedGT with a single group with $n$ clients. The assignment matrix for FL with full secure aggregation is therefore a $1 \times n$ matrix with all ones. For $n = 3$ clients,

$$\boldsymbol{A}_{\mathsf{SA}} = \begin{pmatrix} 1 & 1 & 1 \end{pmatrix} .$$

In Fig. 7a, we depict the trellis representation of the assignment matrix $\boldsymbol{A}_{\mathsf{SA}}$. Note that the non-zero state at depth $\ell = n$ has many incoming edges. This way, even if the group test (i.e., the test on the aggregate) is noiseless, a positive test $\boldsymbol{t} = (1)$ cannot identify the malicious clients, as every client is equally probable to be malicious. Hence, secure aggregation provides privacy at the expense of security.

*Example* 4 (Vanilla federated learning). In vanilla FL, the server has access to all the individual models of the clients and it can perform tests on the clients' individual (not aggregated) models. Therefore, the assignment matrix for vanilla FL is an identity matrix. For $n = 3$,

$$\boldsymbol{A}_{\mathsf{VFL}} = \begin{pmatrix} 1 & 0 & 0 \\ 0 & 1 & 0 \\ 0 & 0 & 1 \end{pmatrix} .$$

The trellis representation of $\boldsymbol{A}_{\mathsf{VFL}}$ is depicted in Fig. 7b. For all trellis depths $0 \leq \ell \leq n$, there is only one incoming edge per state. Therefore, the decoding performed in the trellis is relatively easy, and perfect identification of malicious clients can be achieved provided that the tests are noiseless. Assume that the tests are noiseless and the test outcome is $\boldsymbol{t} \neq \boldsymbol{0}$. Then, from the trellis in Fig. 7b, there is a single path that connects the state $\boldsymbol{t}$ (in decimal) at depth $\ell = 3$ with state 0 at depth $\ell = 0$, thus enabling perfect identification of the malicious clients.

# E  PROOF OF PROPOSITION 1

We re-state the proposition for the reader's convenience.

**Proposition.** *Let the assignment of clients to test groups be defined by assignment matrix $\boldsymbol{A}$ and let $r$ be the smallest non-zero Hamming weight of the vectors in the row span of $\boldsymbol{A}$ (in the coding theory jargon, the minimum Hamming distance of the code generated by $\boldsymbol{A}$ as its generator matrix). Then FedGT achieves the same privacy as a secure aggregation scheme with $r < n$ clients.*

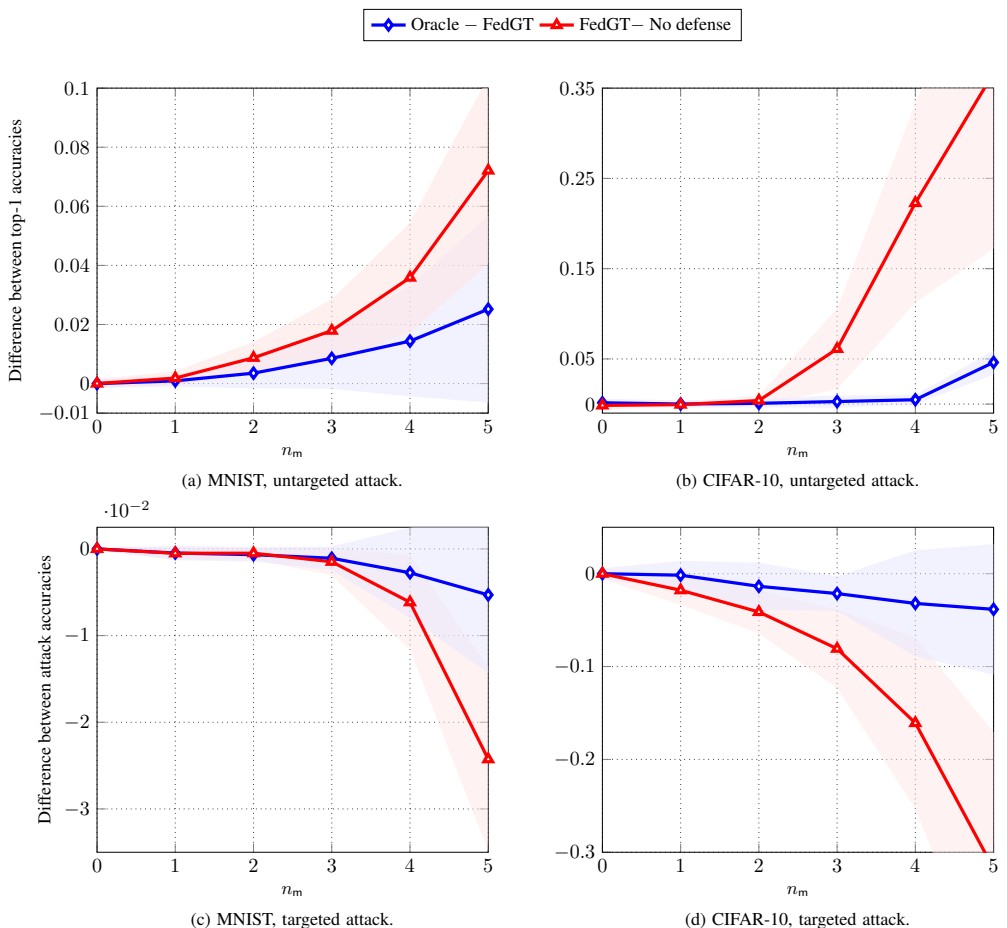

Figure 8: The mean end accuracy difference between FedGT and oracle and no defense. The opaque areas represent the confidence interval ($\pm$ one standard deviation from the mean).

*Proof.* Due to the overlapping groups arising from matrix $\boldsymbol{A}$, there might exist a vector $\boldsymbol{b} \in \mathbb{R}^m$ such that $\sum_{i=1}^m b_i \boldsymbol{u}_i = \boldsymbol{c}_j$ for some $j \in [n]$, or equivalently $\boldsymbol{b}\boldsymbol{A} = \boldsymbol{e}_j$, where $\boldsymbol{e}_j$ is the $j$-th unit vector. This will occur if $\boldsymbol{e}_j \in \mathsf{Sp}_\mathrm{r}(\boldsymbol{A})$, where $\mathsf{Sp}_\mathrm{r}(\boldsymbol{A})$ is the row span of $\boldsymbol{A}$. Generally speaking, for a subset $\mathcal{R} \subset [n]$ of cardinality $r$, if $\sum_{\iota \in \mathcal{R}} f_\iota \boldsymbol{e}_\iota \in \mathsf{Sp}_\mathrm{r}(\boldsymbol{A})$ where $f_\iota \neq 0$, there exists a vector $\boldsymbol{b}'$ such that $\sum_{i=1}^m b'_i \boldsymbol{u}_i = \sum_{\iota \in \mathcal{R}} f_\iota \boldsymbol{c}_\iota$. Thus, we conclude that FedGT achieves the same privacy as a secure aggregation scheme with $r < n$ clients, where $r$ is the smallest non-zero cardinality of the subset $\mathcal{R}$. In other words, $r$ is the smallest non-zero Hamming weight of the vectors in the row span of $\boldsymbol{A}$. $\quad\square$

## F STATISTICAL VARIATIONS OF THE NUMERICAL RESULTS

The results shown in Figs. 1, 3-6 are obtained from experiments using a Monte-Carlo strategy, i.e., the experiments are conducted more than once and the mean is presented. More specifically, the experiments over the MNIST dataset are performed 100 times, while the ones for CIFAR-10 are performed 5 times. We choose a smaller number of Monte-Carlo iterations for CIFAR-10 due to the higher training complexity of Resnet-18. Due to heterogeneity in the data points of the ISIC2019 dataset, we performed a single round of experiments on this dataset and consequently omit it from the present analysis.

An essential aspect of the Monte-Carlo-based experiments is that, due to the law of large numbers, the result averaged over a large number of iterations approaches the real outcome of the results. However, running FL experiments many times quickly becomes unfeasible, thus limiting the maximum number of Monte-Carlo iterations we can perform.

To reduce the simulation noise due to a small number of Monte-Carlo iterations, we performed our experiments using a fixed seed for experiments with FedGT, oracle, no defense, and RFA. Note that each Monte-Carlo iteration has its own unique seed, which ensures different realizations of the defective vector $d$ and data partition for the clients and the server validation dataset. However, for a given Monte-Carlo iteration, all four strategies (FedGT, oracle, no defense, and RFA) are evaluated on the same realization, thus making the comparison between the strategies fair.

In Fig. 8, we show the statistical variation of the numerical results presented in Fig. 6 for FedGT, oracle, and no defense. Specifically, since we have dependent experiments across strategies (due to the fixed seeding), we compute the differences in accuracies (top-1 accuracy for the untargeted attack and attack accuracy for the targeted attack) at the last communication round (10 for experiments over MNIST and 50 for CIFAR-10) between oracle and FedGT, and FedGT and no defense for every Monte-Carlo iteration. In Fig. 8, the mean of the differences and the confidence interval ($\pm$ one standard deviation) are illustrated.

Figs. 8a and 8b show the differences in top-1 accuracy evaluated in the last communication round for experiments over MNIST and CIFAR-10 datasets, respectively. We see that for low prevalences ($n_{\mathsf{m}} \leq 2$), the three strategies perform more or less similar regardless of the dataset. Moreover, the differences in top-1 accuracy are concentrated in the mean. However, for MNIST and $n_{\mathsf{m}} \in \{3, 4, 5\}$, in average oracle is better than FedGT, and FedGT performs significantly better than no defense. For $n_{\mathsf{m}} = 5$, FedGT has a performance gain from $4\%$ to $10\%$ compared to no defense, but it can have a performance loss of $5\%$ compared to oracle. However, the performance loss compared to oracle can also be negligible for some Monte-Carlo iterations even for $n_{\mathsf{m}} = 5$. For the experiments over CIFAR-10 (see Fig. 8b), and $n_{\mathsf{m}} \in \{3, 4, 5\}$, there is a positive performance gain of FedGT that can go up to $40\%$. The performance loss of FedGT compared to oracle seems very concentrated around the mean and goes at maximum $5\%$ for $n_{\mathsf{m}} = 5$.

In Figs. 8c and 8d, we show the statistical variations of the results shown in Fig. 1 for targeted attack experiments over MNIST and CIFAR-10 datasets, respectively. Note that a higher attack accuracy means success for malicious clients. We conclude that for experiments over MNIST, it becomes beneficial to use FedGT for $n_{\mathsf{m}} \in \{4, 5\}$, even though the performance gain and loss is minor. It is important to note that for lower prevalence $n_{\mathsf{m}} \leq 3$, FedGT does not worsen the performance. On the other hand, experiments over CIFAR-10 dataset (see Fig. 8d) show that FedGT has a significantly better performance compared to the no defense strategy for prevalences $n_{\mathsf{m}} \in \{2, 3, 4, 5\}$. For example, for $n_{\mathsf{m}} = 4$, FedGT has a smaller attack accuracy in the range of $7\%$ to $25\%$. Fig. 8d also shows that, on average, FedGT performs closely to the oracle strategy, and it can deteriorate the attack accuracy by at most $10\%$, for $n_{\mathsf{m}} \in \{4, 5\}$, compared to oracle.

## G  Mismatched decoding

As described in Section 5.2, the forward-backward decoding algorithm requires the knowledge of the prevalence $\delta$ and a probabilistic model $Q(t|s)$ for the noisy test. The results shown in Figs. 1, 3-6, are obtained by feeding to the decoder the prevalence $\delta = n_{\mathsf{m}}/n$ (except for $n_{\mathsf{m}} = 0$, where $\tilde{\delta} = 0.1$ is used), which requires the prior knowledge of $n_{\mathsf{m}}$. Moreover, we have considered a binary symmetric channel $Q(t|s)$ with crossover probability $p = 0.05$ as the noise test model. Although the forward-backward decoder is compatible with asymmetric modeling of the noisy tests, in this paper we utilize a symmetric probabilistic model.

In Fig. 9, we present experimental results to show the robustness of FedGT to a mismatched prevalence $\tilde{\delta}$ (i.e., a value that does not correspond to the true $\delta$) and for other choices of the crossover probability $p$. We have considered an untargeted attack and followed the same testing strategy described in Section 6.2. The hyperparameters are the same as in Section 6. The experiments were conducted over MNIST and CIFAR-10 datasets for the scenario $n_{\mathsf{m}} = 3$ and $n = 15$. We pick mismatched values for the prevalence $\tilde{\delta} = \{0.05, 0.15, 0.25, 0.35\}$ and we run experiments for crossover probabilities $p \in \{0, 0.05, 0.1, 0.2\}$. Note that the particular case $p = 0$ means that the testing strategy is assumed to be noiseless. We run the experiments using a Monte-Carlo approach, where we use 30 Monte-Carlo iterations for MNIST and 5 for experiments over CIFAR and the results are averaged. We sweep the decoding threshold $\Lambda \in \{0.1, 0.2, \ldots, 0.9\}$ for experiments over MNIST and $\Lambda \in \{0.1, 0.3, \ldots, 0.9\}$ for CIFAR. In the plots in Fig. 9, the best lambda is picked and presented for each choice of $\tilde{\delta}$ and $p$.

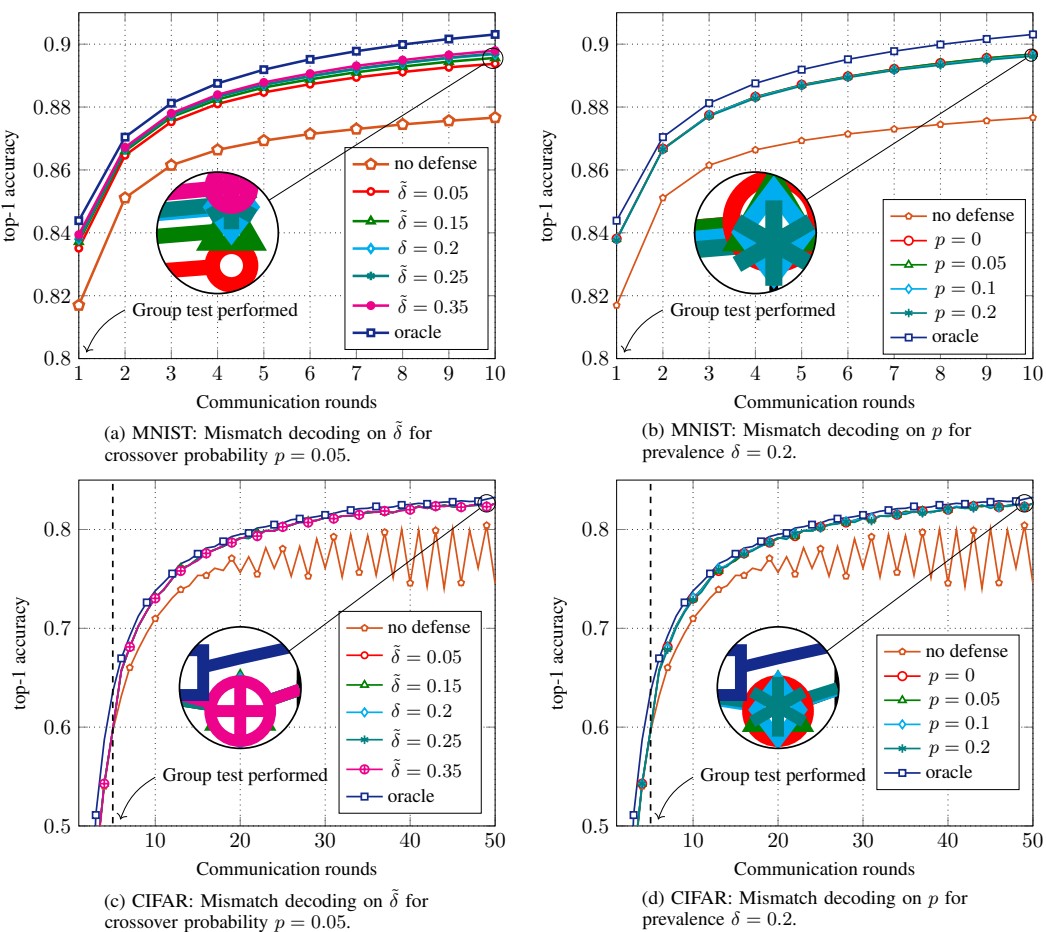

Figure 9: Experimental results using mismatch prevalence $\tilde{\delta}$ and crossover probability of the noisy test model $p$.

The plots in Figs. 9a and 9b, show the impact of using a mismatched prevalence and different crossover probabilities, respectively, in the top-1 accuracy for the MNIST dataset. In Fig. 9a, we can see a slight difference in the top-1 accuracy due to the mismatched prevalence, but the difference is negligible. The highest accuracy is achieved for $\tilde{\delta} \in \{0.2, 0.25, 0.35\}$ (which are within $0.001\%$ in final top-1 accuracy), and the difference between the highest accuracy and the lowest (due to the mismatch) is $0.005\%$. The experiments over different crossover probabilities $p$ show that the effect in accuracy is also negligible, see Fig. 9b. The difference of the (final) top-1 accuracy for different values of $p$ is less than $0.001\%$. Hence, we conclude that over MNIST, FedGT is robust to a mismatch in prevalence and crossover probability.

In Fig. 9c, we plot the results obtained by using a mismatched prevalence $\tilde{\delta}$. Interestingly, the mismatch in prevalence results in little to no effect on experiments over CIFAR-10. From our experiments, we observed that the difference in the performance of FedGT is at most $0.0001\%$ by using a mismatched prevalence. Fig. 9d, where the results for different crossover probability $p$ are shown, conveys the same message. The difference in end accuracy using different values of $p$ is negligible, amounting to a value of $0.0003\%$ at most. Therefore, we conclude that FedGT is robust to a mismatch in prevalence and $p$ also for experiments over the CIFAR-10 dataset. However, we advise not to consider extreme cases of mismatches in $\delta$ and $p$, for example, picking both of them $\tilde{\delta} = p = 0.5$ will force the decoder to choose a random path in the trellis as the decoding output.

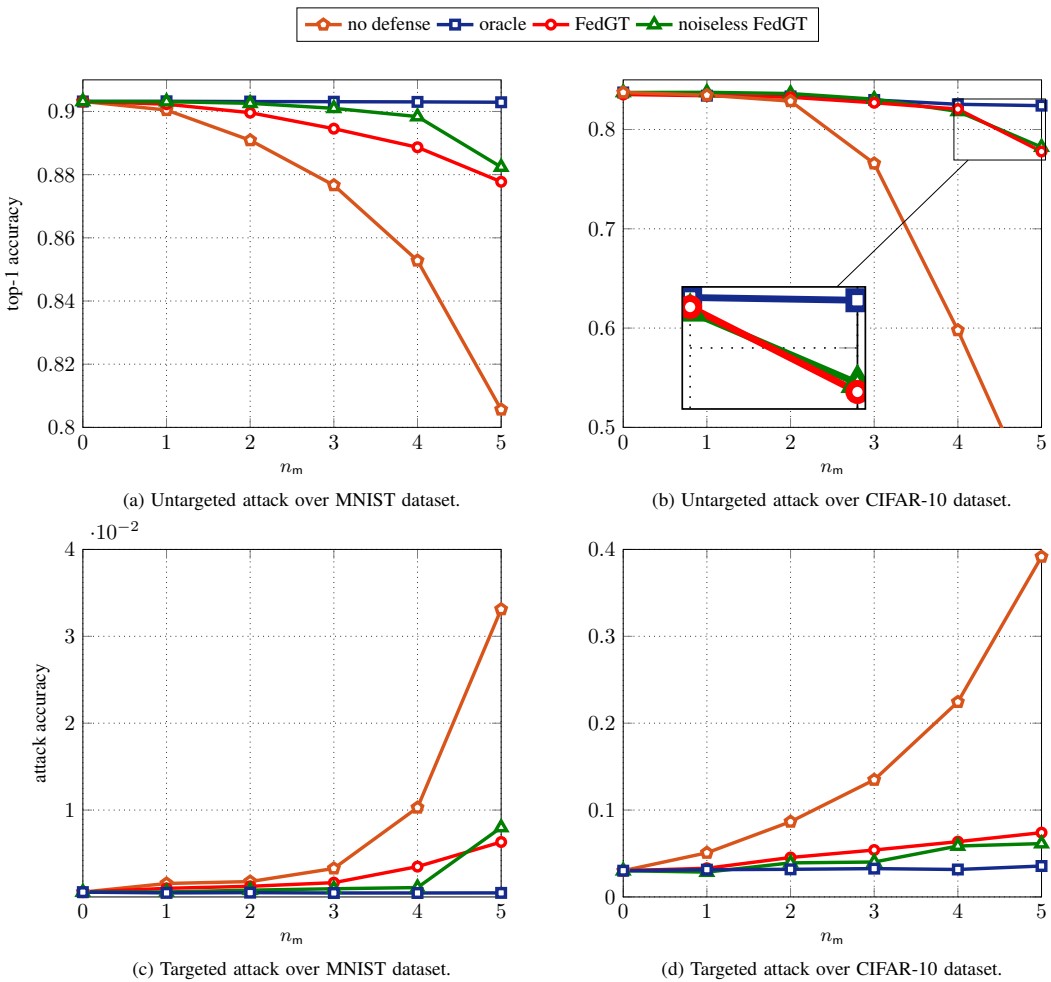

Figure 10: Performance of FedGT with perfect (noiseless) testing for experiments with untargeted (first row) and targeted (second row) attacks.

## H   TESTING STRATEGIES

In this work, we provided experiments that assume the presence of a small validation dataset at the server and employ very simple tests. However, FedGT is not restricted to this assumption and is, in fact, test-agnostic. Devising more sophisticated testing strategies may enhance the performance of FedGT. In general, the performance will depend on the noisiness of the tests: less noisy tests will in general yield better results.

Moreover, we considered the *weakest* version of FedGT, in the sense that we perform group testing only at a single communication round. The protocol can be modified such that testing is performed in multiple rounds and, further, the previous decoding outcome may be used for future tests and decoding. The testing strategy can also be a composite of multiple different tests and/or be performed by tracking the aggregated models of earlier communication rounds and comparing them. This way, FedGT can provide resiliency against a smarter attack that switches the status (benign or malicious) between rounds. FedGT can be modified such that the groups are randomized in different rounds. However, extra leakage might occur due to the re-grouping and further analysis in this scenario is necessary.

In Fig. 10, we plot the performance of FedGT with a perfect testing strategy, i.e., a test such that $t = s$, denoted as noiseless FedGT. To obtain these results, we feed the decoder the correct test outcome, $t = s$, prevalence $\delta = n_{\mathrm{m}}/n$, and $p = 0$. We follow the same strategy as for FedGT, if

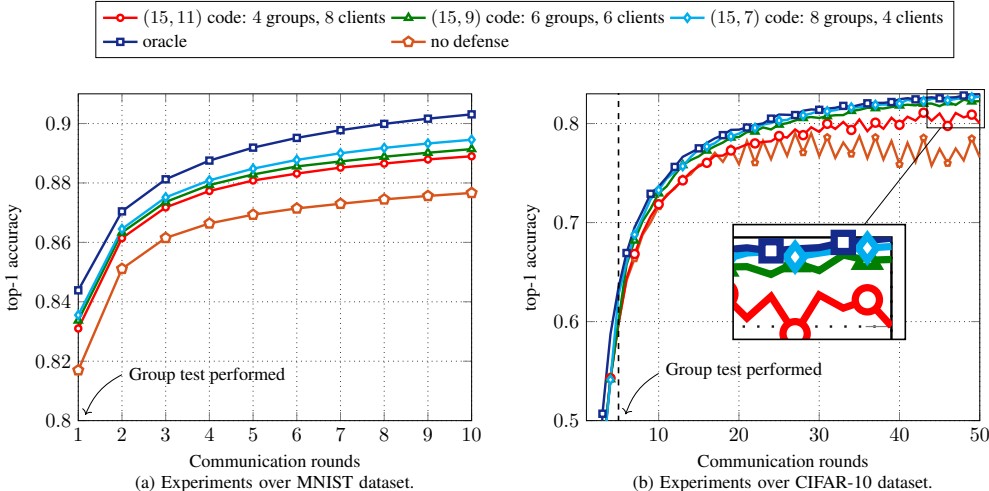

Figure 11: We present experimental results where FedGT uses different codes for grouping the clients. The codes are picked to have the same length 15 and yield different numbers of groups. More groups are translated into less privacy for the clients and vice-versa.

$\hat{d} = 1$, the training proceeds without a defense, i.e., we fix $\hat{d} = 0$. The plots shown in Figs. 10a and 10b show the top-1 accuracy as a function of $n_m$, similar to the plots presented in Figs. 6d and 6e, with the addition of the performance of noiseless FedGT. Similarly, Figs. 10c and 10d show the comparison of noiseless FedGT with the curves already presented in Figs. 6a and 6b. In Fig. 10a, we see that there is a gap between FedGT and noiseless FedGT. In general, we see from Fig. 10 that FedGT and noiseless FedGT have similar performance. However, this is mostly happening because noiseless FedGT has higher chances of obtaining $\hat{d} = 1$ compared to FedGT (due to the testing strategy). Our experimental results show that if there exists at least one group that does not contain any malicious clients, then noiseless FedGT outperforms FedGT. Hence, our testing strategy is noisy and better tests may be envisaged.

## I PRIVACY-SECURITY TRADE-OFF

Throughout the paper, we have considered a scenario with $n = 15$ clients and the parity-check matrix of a $(15, 7)$ BCH code of length 15 and dimension 7 as the assignment matrix. This corresponds to a FedGT scheme with 15 clients and 8 groups, each with 4 clients, and the privacy achieved is the same as secure aggregation with 4 clients. However, there are other ways of grouping the clients, where the number of groups can be smaller and the number of clients per group larger, yielding a trade-off between privacy and security. To highlight this trade-off, here we consider two other assignment matrices obtained by taking the parity-check matrices of cyclic codes of length 15. More specifically, we choose two other cyclic codes: A $(15, 9)$ code of length 15 and dimension 9 and a $(15, 11)$ code of length 15 and dimension 11. The $(15, 9)$ code results into 6 groups of 6 clients each and a level of privacy equal to that of secure aggregation with 6 clients. The $(15, 11)$ code results into 4 groups of 8 clients each and a level of privacy equal to that of secure aggregation with 8 clients. As expected, reducing the number of groups increases the clients' privacy. This occurs since the dual of the code used for grouping has a lower dimension and hence enables a higher minimum Hamming distance.

In Fig. 11, we present the impact on the accuracy when using assignment matrices with a different number of groups and number of clients per group, highlighting the trade-off between privacy and security. The experiments are run for the MNIST and CIFAR datasets (for $n_m = 3$, untargeted attack). The results are shown in Figs. 11a and 11b, respectively. For both datasets, we observe that increasing the number of groups (equivalently, lowering the code rate) yields a higher accuracy. This is expected, as the code is stronger and can correct more errors. However, the higher accuracy comes at the expense of a lower privacy, as discussed above. In contrast, decreasing the number of groups (i.e., increasing the code rate) increases privacy, but reduces the learning accuracy. The latter effect

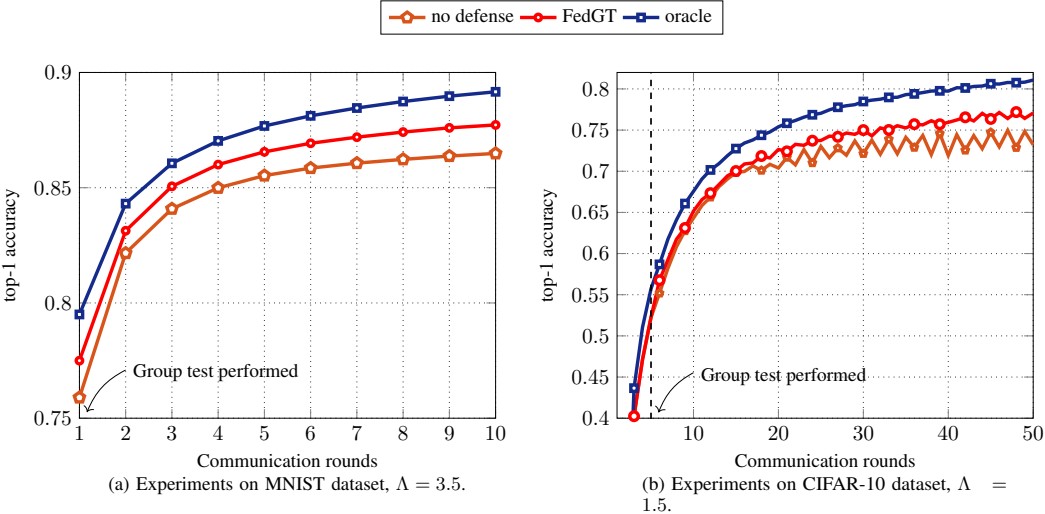

Figure 12: Experimental results for a federated learning with $n = 31$ clients out of which $n_\mathrm{m} = 6$ of them are malicious. The malicious clients follow an untargeted attack strategy. We show the performance of FedGT and we compare it with the no defense and oracle strategy.

occurs since the capability of identifying malicious clients deteriorates due to the use of a weaker code.

## J  FEDERATED LEARNING WITH MORE CLIENTS

Throughout this paper, we have restricted the focus of the experiments to a cross-silo FL scenario with 15 clients. Here, we investigate the performance of FedGT for another cross-silo FL scenario with more clients, specifically $n = 31$ clients. To apply FedGT to this scenario, we choose as the assignment matrix the parity-check matrix of a $(31, 21)$ BCH code of length 31 and dimension 21, resulting in 10 groups, each containing 12 clients. The dual of this BCH code has minimum Hamming distance 12, thus FedGT preserves the same clients' privacy of secure aggregation with 12 clients, which is almost $40\%$ of the total number of clients.

We investigate a scenario where $n_\mathrm{m} = 6$ malicious clients are present and they perform an untargeted attack, as explained in Section 6.2. The results are shown in Fig. 12. We conduct experiments over the MNIST and CIFAR-10 datasets, where the hyperparameters considered are specified in Section 6. The results are shown in Figs. 12a and 12b, respectively. We run the experiments for $\Lambda \in \{1, 1.5, \dots, 3.5\}$ and plot the results for the best $\Lambda$. We compare the performance of FedGT with that of no defense and the oracle strategy.

From Fig. 12, we observe that regardless of the dataset, FedGT has a better performance compared to no defense. However, there is a non-negligible gap between the performance of FedGT and the oracle strategy. This is expected since the number of clients per group is relatively high, resulting in a large overlap between groups, which ultimately makes it challenging to render groups containing only benign clients. An improved accuracy (at the expense of lower privacy) can be achieved by considering a lower rate code (i.e., with more groups). In this case, however, optimal decoding via the forward-backward algorithm becomes quickly infeasible and one needs to rely on suboptimal decoding strategies (see Appendix L, Item 3). Furthermore, better performance can be achieved by improving the testing strategy, as discussed in Appendix H. Improving the testing strategy seems crucial for larger scenarios (larger $n$).

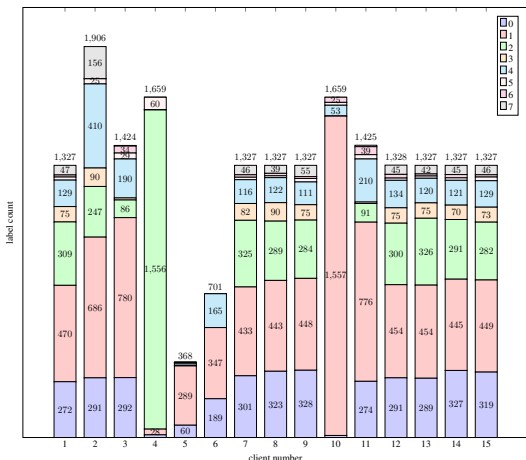

Figure 13: Client data partitions for experiments over the ISIC2019 dataset
.

## K    DETAILS OF THE ISIC2019 EXPERIMENT

The ISIC2019 dataset (Codella et al., 2018) is a public dataset consisting of images of various skin lesion types, including malignant melanomas and benign moles, used for research in dermatology and skin cancer detection. We use the ISIC2019 dataset and follow (Ogier du Terrail et al., 2022) by first restricting our usage to 23247 data samples (out of 25331 entries) from the public training set due to metadata availability and then preprocessing by resizing the shorter side to 224 pixels and by normalizing the contrast and brightness of the images. Next, we randomly divide the data into a test and a training dataset of size 3388 and 19859, respectively. The server validation set is created by randomly sampling 100 data entries from the training set. Next, as in (Ogier du Terrail et al., 2022), the remaining 19759 training samples are partitioned into 6 partitions with respect to the image acquisition system used. The 6 partitions are then split into 15 partitions by iteratively splitting the largest partition in half. This procedure results in 15 partitions with heterogeneity in number of data samples and label distribution, as seen in Fig. 13, and in the feature distribution due to different acquisition systems, as visualized in (Ogier du Terrail et al., 2022, Fig. 1.f).

Due to the large imbalance in the dataset (label 1 corresponds to 48.7% whereas label 5 and 6 are represented by about 1% of the entries), the focal loss is used in the training (Lin et al., 2017) and we use the balanced accuracy to assess the performance of the trained network. Furthermore, to encourage generalization during training, we follow (Ogier du Terrail et al., 2022, Appendix H) and apply random augmentations to the training data.

Finally, the heterogeneous data partitioning causes the choice of malicious clients to significantly impact the outcome of the experiment. For this reason, we let the set of malicious clients $\mathcal{M}_i \subset \mathcal{M}_j$ for $j > i$ to ensure that results across different values of $n_{\mathrm{m}}$ are comparable. In particular, we let $\mathcal{M}_1 = \{2\}$, $\mathcal{M}_2 = \{2, 7\}$, $\mathcal{M}_3 = \{2, 7, 9\}$, $\mathcal{M}_4 = \{2, 7, 9, 10\}$, $\mathcal{M}_5 = \{2, 7, 9, 10, 15\}$, respectively.

## L    LIMITATIONS

In this section, we identify and discuss some limitations of the proposed framework, FedGT.

- **Dependence on a good test:** Any group testing framework relies on an appropriate test. FedGT, as a group testing-based framework, does not constitute an exception. Thus, the performance of FedGT will heavily depend on the choice of the testing strategy adopted. Ideally, the testing strategy should be noiseless. FedGT proposes that testing is performed on aggregated models and not on individual clients' models (to preserve privacy). However, finding good testing strategies over aggregated models is an active research area and further investigations are necessary. Furthermore, in our paper we considered a simple testing

strategy (measuring the accuracy or the recall once after a number of communication rounds). More sophisticated testing strategies could be considered, potentially leading to improved results, especially for large $n$.

- **Validation dataset:** The tests in this paper hinge upon the use of a relatively small validation dataset. Generally, assuming the presence of a validation dataset on the server constitutes a substantial assumption, let alone a dataset that mirrors the clients' distribution closely. Nevertheless, the FedGT framework's versatility accommodates a wide range of tests, including those that do not rely on a validation set, e.g., tests based on the spectral decomposition of models (Tolpegin et al., 2020). This feature allows for the incorporation of tests initially designed for non-private defenses into the FedGT framework, thereby ensuring client privacy is upheld.

- **Scalability to a cross-device FL scenario with a very large number of clients:** In this work, our primary focus lies in the context of a cross-silo scenario, characterized by a limited number of clients (up to 100) (Shejwalkar et al., 2022). This allows us to compute exact a posteriori LLRs using the forward-backward algorithm. For a cross-device scenario, wherein the number of clients is significantly larger (resulting in a very large matrix $A$), the exact computation of the LLRs becomes infeasible for full client participation. Although FedGT is general and extends to the cross-device FL scenario, to tackle the decoding complexity challenge associated with a very large assignment matrix $A$, it becomes imperative to utilize a matrix $A$ corresponding to a sparse-graph code (e.g., a low-density parity-check (LDPC) code or a generalized LDPC code) and employ (suboptimal) message-passing decoding techniques such as belief propagation (Kschischang et al., 2001). Message-passing decoding for codes on graphs is based on the exchange of messages over the edges of the bipartite graph representation of the code (in our scenario, the bipartite graph corresponding to $A$, see Fig. 2a). Message-passing decoding is known to perform very well for large sparse matrices. Furthermore, LDPC and generalized LDPC codes have been shown to yield excellent performance for group testing (Mashauri et al., 2023; Lee et al., 2019; Karimi et al., 2019). In our future work, we will consider FedGT based on LDPC codes and generalized LDPC codes for the identification of malicious clients in a cross-device scenario.

  For cross-device FL, it is common to sample different clients over the communication rounds. If the number of sampled clients is $< 100$, FedGT can be applied in every round to detect malicious clients. However, the privacy impact on clients participating in multiple rounds and in different groups is currently unknown. The cross-device scenario is a natural extension to the current work that we will pursue in the future.

- **Beyond data poisoning:** The FedGT framework organizes clients into several subgroups, within which secure aggregation is practiced. It is crucial for the integrity of each subgroup that clients contribute the same local model to all subgroups they participate in—this is inherently true for data-poisoning attacks. A model poisoning attack, where the malicious clients upload arbitrary model parameters, may leverage the sub-grouping to its advantage by submitting benign models during the testing stage and a malicious model during the subsequent step, effectively becoming invisible to FedGT. To address this issue, one may utilize cryptographic techniques to ensure model consistency from each client over subgroups (and the final aggregation of benign clients) at the expense of an increase in communication cost.

- **Test timing:** The FedGT framework permits client testing at any given round. However, it is important to note that each round of testing inevitably leads to an increase in communication overhead due to the secure aggregation conducted across subgroups. It may be prudent to conduct tests periodically to ensure the integrity of the training process. The optimal timing of these tests might vary depending on the testing strategy employed. For instance, with simpler tests as considered in this paper, testing should ideally occur once the local models have begun to extract meaningful patterns from the data. For the MNIST dataset, this learning phase begins as early as the first communication round, while for CIFAR-10 and ISIC2019 datasets, it takes place after a few communication rounds.

