# OpenReview forum: "FedGT: Identification of Malicious Clients in Federated Learning with Secure Aggregation"
_ICLR.cc/2024/Conference — Submitted to ICLR 2024_

### Official Review · Reviewer_CiYQ · 2023-10-18

**Soundness:** 2 fair
**Presentation:** 3 good
**Contribution:** 1 poor
**Rating:** 3
**Confidence:** 3

**Summary:**

The paper proposes a new scheme for defending against poisoning attacks in cross-silo FL under secure aggregation. The scheme uses optimal coding techniques to split the federated learning clients into possibly overlapping groups. Each group's models are securely aggregated into a single model the server observes. Each group's aggregated model is tested by the server separately on auxiliary data for maliciousness. This allows the server to detect the presence of malicious or severely out-of-distribution clients in each group and, from that, deduce the identity of the malicious clients based on their group participation patterns cleverly chosen by the server. Finally, the server removes the perceived malicious clients from the federated process. The scheme tradeoffs privacy and security by allowing different sizes of groups to be chosen by the server, which, thanks to the optimal coding scheme used, also corresponds to the effective secure aggregation size of the FL process. The authors show experiments on different datasets and models, demonstrating that the scheme can be effective at defending against malicious clients while ensuring the privacy of benign clients.

**Strengths:**

- Solves an important problem
- The proposed method is an interesting application of group testing theory.
- Encouraging results that such a method can work in this setting despite the simplistic instantiation of it
- The method can work even when applied only at a single communication round.
- Interesting connection between coding theory and privacy of secure aggregation

**Weaknesses:**

- **Structure and Novelty of the submission:**
There are three major components to this paper. First, the paper suggests the use of group-testing strategies to determine which clients are malicious in the context of the cross-silo FL setup with secure aggregation. In this context, the authors prove how overlapping test groups affect the overall client privacy, as secure aggregation is not commonly applied over overlapping groups. This part of the paper is novel and constitutes the biggest technical contribution of the paper. Then, the authors use optimal codes to determine an efficient way to group clients such that a chosen level of privacy is ensured while the decoding ( aka malicious client identification ) can work well. This is mostly based on prior work from coding theory and contains little to no technical novelty. Finally, the paper then applies the optimal detection strategy from [1] without any changes to the problem of malicious client identification. The paper does not give proper credit to [1], given that it essentially just applies the algorithm presented there and instead re-derives most of the work in Section 5. To this end, there is little to no novel technical contribution in that part of the paper either. One area where the paper could have expanded on [1] and could have provided a real technical contribution is instantiating the many choices of [1] to the FL problem considered. Unfortunately, the paper does very little of that as it takes the absolute simplest choice for a test, models the threshold $\Lambda$ as a tunable hyperparameter ( pretty much independent of the proposed FA/MD trade-off ), assumes $\delta$ is known ( something that even the authors agree is not a reasonable assumption ), and use embarrassingly simple $Q(t|s)$ model. All in all, therefore, the paper provides very little novel technical contribution, which is sad, as the idea of applying group testing to this problem seems interesting and promising.
Structurally, the paper wastes Section 5 by re-deriving [1] without giving [1] the proper credit and also compromising the explanation of the decoding, as the space is not enough for a full explanation of the decoding method. This space would have been much better spent on explaining the instantiation of [1] to the federated setting, e.g., what coding scheme is used, what test is used, how is $Q(t|s)$ chosen, etc., which is currently done all over the place. Further, it would have been good if the paper explained the properties of this instantiation and why it makes sense in the federated case. For example, the paper could have explained what the chosen coding strategies guarantee in the context of the FL problem being solved, what their communication complexity is, and how reasonable a test based on additional validation data and known attack parameters ( such as the targetted classes and number of malicious clients ) is.
- **The defense relies on unreasonable assumptions and introduces too many hyperparameters:**
While the experiments provided in the paper suggest the proposed group testing strategies have the potential to constitute a good defense, the particular instantiation is unsatisfactory.
In particular, in the main paper, the authors assume unrealistic knowledge by the defender, such as known $\delta$ and known target class in the targetted version of the attack. While $\delta$ is shown to be not that important in Figure 9, wrong $\delta$ should be the default setting in which all experiments are carried out, as it is the only way to carry out the attack in the first place. Providing experiments with known $\delta$ is fine for ablation purposes but nothing more. Moreover, the known target is also a big assumption, as if the test is carried out for multiple targets, this can introduce additional noise in the test and lower the presented results.
Further, the added hyperparameters $\Lambda$, $p$, and $\rho$ are hard to tune in practice. In particular, choosing $\Lambda$ heuristically depending on the setting, is hard and only possible through several re-trainings of the model. This is caused by the author's choice to tune $\Lambda$, which is an uninterpretable parameter in $(-\infty,\infty)$ to which the results are very sensitive, instead of $\Delta'$, which can be interpreted as a tradeoff between FA and MD. Further, I find the chosen model of $Q(t|s)$ which only depends on a single $p$ (the probability of error), too simplistic and hard to guess. I would suggest the authors instead at least choose $Q(t|s)$ as a probability distribution of the hamming distance between $t$ and $s$. This will at least account for the fact that individual tests can fail separately. Finally, $\rho$ is likely to be dependent on the heterogeneity level and complexity of the problem and might also be hard to tune.
- **Current experimental results are not strong enough:**
Despite the multiple hyperparameters introduced by the authors, the method still produces results that, in many cases, do not beat the baseline the authors themselves provide( [2] ). This is especially visible in Figures 4 and 5 in the Appendix, where the authors' method consistently performs worse in simple settings like MNIST. It seems that the proposed model, as it is, is mostly capable of beating [2] in heterogeneous settings where targetted attacks with known targets are applied. While this is an important setting that is worth solving separately, this is a clear drawback of the method that is not discussed by the authors. If this is where the method really works better than alternatives, the whole paper needs to focus on this setting and explain why this setting is favorable to their method ( or unfavorable for other methods ). Further, to allow realistic apples-to-apples comparisons, as mentioned above, authors should remove the known target and $\delta$ assumptions (which I assume are not needed by [2]) and discuss how many clients are being securely aggregated by [2]. If the experiments for [2] aggregate all clients ( which I suspect is the case ), then the numbers are even worse for the proposed method, as [2] is in the "most private" setting where security is harder to achieve and yet still beats the proposed method regularly.
Finally, another concern regarding the results is the fact the method seems to affect more clean accuracy compared to [2] and even the no defense ( according to Tables 1-3 ). This might suggest that good defense properties demonstrated in the paper are currently achieved on the back of too-high FA rates, tanking the clean performance.
- **Runtime concerns:**
The authors say in the main paper repeatedly they focus on the cross-silo FL setting which they define to be up to 100 devices. Yet, in their experiments, the authors only use 15, and in the scaling experiment in Appendix J, they claim they do not even scale to 31 when some more precise encodings are used. This needs to be explicitly discussed in the main paper, but also severely jeopardizes the applicability of the proposed algorithm.
- **Other:**
1. The discussion on communication complexity needs to be at least partially moved to the main paper. The authors keep claiming their communication complexity as an important contribution, but unless one reads the appendix, they do not even know what the communication complexity of this and baseline models are.
2. In all experiments, provide information on what the FA and MD rates are. Further, provide experiments where you change $\Lambda$ and show how this affects FA and MD rates and how it affects the attacker's accuracy and the model's clean accuracy.
3. Make explicit in **the main paper** that the current methodology works only for data poisoning and not model poisoning. ( Essentially bring the "**Beyond data poisoning**" point from Appendix L to the main paper )
4. If the authors plan on keeping Sec. 5, which I do not recommend, they should at least provide some justification for the forward and backward formulas.
- **Nits:**
1. The last paragraph on page 3 does not explain when a test is positive ( i.e. is it when the group has a malicious client or when there isn't ). Add this to the definition
2. The $s=d \vee A^T$ notation, when first introduced, is confusing if the reader is unfamiliar with the group testing literature. Maybe explain that this is matrix multiplication where the dot product uses "ands" for multiplication and "ors" for additions.

**Questions:**

- Can the authors provide an additional experiment where they change $\Lambda$ and show how this affects the FA and MD rates and how it affects the attacker's accuracy and the model's clean accuracy? Which is more detrimental for the performance of FedGT - high FA or high MD ratios?
- Can the authors explain why the CIFAR10 attack success goes down even without defense in Figure 4 ?
- Can the authors provide a linear scale version of Figure 4? Log scale in the attacker's accuracy both makes it hard to read and also obfuscates the results, which are not naturally log distributed.
- Can the authors provide experiments where the defender does not know the targets for the targeted attacks?
- Can the authors explain how many clients are securely aggregated by [2] in their experiments?
- Can the authors provide runtimes of the method for different numbers of clients and encodings? Can the authors provide a worst-case or average-case runtime analysis (in terms of big-O notation)?

All in all, this paper has the potential to introduce a very interesting defense mechanism with good performance and interesting properties. However, it does not do any of this. It does not properly instantiate the algorithms used in the federated setting, as it chooses super basic tests and sets everything else to impossible to obtain in practice hyperparameters. Possibly due to this, the results on the defense side are underwhelming, even when the comparison is done in a very favorable way to the proposed method by giving more information to the defender than typically assumed while also comparing to a method that cannot benefit from the trade-off described in this paper in terms of security-vs-privacy. Further, the method seems to be very computationally expensive to the server in cases where more than a handful of clients are available, and the clean accuracy is sometimes severely penalized possibly due to a high FA ratio.


[1] Gianluigi Liva, Enrico Paolini, and Marco Chiani. Optimum detection of defective elements in non-adaptive group testing. In Annu. Conf. Information Sciences and Systems (CISS), Baltimore, MD, 2021.
[2] Krishna Pillutla, Sham M. Kakade, and Zaid Harchaoui. Robust aggregation for federated learning.
IEEE Transactions on Signal Processing, 70:1142–1154, 2022.

**Details Of Ethics Concerns:**

There is none.

---

> ### Author Response · Authors · 2023-11-20
> **Response to Reviewer CiYQ**
>
> We would like to thank the reviewer for thoroughly reading the manuscript and for the insightful comments in order to make the paper stronger. Below, you may find the answer to the questions posted.
>
> - **Structure and novelty**: We acknowledge that channel coding and its relation to group testing is not novel. However, as this work is written towards the machine learning community, we deem Section 5 important as it provides the necessary background to the proposed method. Indeed, another reviewer asked for more information on this part. To the authors' knowledge, this paper constitutes the first marriage of group testing and federated learning. This further allows the server to identify malicious clients, something that has previously not been possible. Furthermore, as the reviewer points out, by utilizing the connection to channel coding, we are able to relate the group testing strategy to secure aggregation protocols, something that is also novel.
>
> - **Assumptions and hyperparameters**: We acknowledge that a known target may seem like a big assumption. However, by inspecting the recalls of the different classes in the validation dataset, it turns out to be easy to identify labels that are under attack. In the table below, we illustrate the class recalls for each group over all labels (only one is under attack) for the experiments over CIFAR10 dataset (label 7 is under attack). Note that the groups have different numbers of malicious clients and we denote this by $\tilde{n}_m$ in the table. Hence, we argue that it is easy for the server to simply compute the class recall for each class; if an anomaly is observed in the evaluated recalls, that class is likely under attack. Given the large discrepancy between the class recall of a benign model and a poisoned model, we conduct the experiments under the assumption that the targeted label is easily identified.
>
>     |$\tilde{n}_m$ \ label| 0 | 1 | 2 | 3 | 4 | 5 | 6 | 7 | 8 | 9 |
>     |-----|------|------|------|------|------|------|------|------|------|------|
>     | 1 | 0.55 | 0.17 | 0.33 | 0.6 | 0.71 | 0.33 | 0.85 | 0.0 | 0.91 | 0.56 |
>     | 0 | 0.45 | 0.17 | 0.44 | 0.4 | 0.57 | 0.67 | 0.77 | 0.69 | 1.0 | 0.78 |
>     | 1 | 0.36 | 0.17 | 0.33 | 0.4 | 0.79 | 0.67 | 0.85 | 0.08 | 1.0 | 0.67 |
>     | 0 | 0.64 | 0.33 | 0.33 | 0.4 | 0.71 | 0.56 | 0.69 | 0.54 | 1.0 | 0.56 |
>     | 2 | 0.82 | 0.17 | 0.22 | 0.4 | 0.86 | 0.67 | 0.77 | 0.0 | 0.82 | 0.56 |
>     | 1 | 0.64 | 0.17 | 0.33 | 0.4 | 0.71 | 0.67 | 0.92 | 0.08 | 1.0 | 0.67 |
>     | 2 | 0.55 | 0.33 | 0.33 | 0.6 | 0.71 | 0.44 | 0.85 | 0.0 | 0.91 | 0.56 |
>     | 1 | 0.45 | 0.33 | 0.44 | 0.6 | 0.64 | 0.67 | 0.77 | 0.0 | 1.0 | 0.67 |
>
> We thank the reviewer for the suggestion of considering a different $Q(\mathbf{t}\lvert \mathbf{s})$---it is indeed an interesting direction. For sake of clarification, we model the noisiness of the test $Q(t_i\lvert s_i)$ as a BSC in order to make it more realistic. For example, an asymmetric binary channel might be hard to motivate in a general real-life problem, even though the decoder we use is capable of handling an asymmetric model. Then, we treat the overall noise of the tests to kick-start the backward recursion in the forward-backward decoder $Q(\mathbf{t}\lvert \mathbf{s})$ as $m$ (size of $\mathbf{t}$) parallel BSC with crossover probability $p$. We understand that a smarter way is to treat $Q(\mathbf{t}\lvert \mathbf{s})$ as a parallel set of several BSCs with crossover probability $p_i$ for $i \in [m]$. Indeed, with our test, one can incorporate this information by investigating the clustering of the group accuracies/source recalls. However, note that the best one can achieve by incorporating this information is to approach the performance of a potential noiseless (genie-aided) testing strategy. However, we observe that FedGT is not performing very far away from its noiseless version (see Figure 10). Therefore, the complexity needed to choose suitable $p_i$'s on the fly does not pay off in terms of performance. Another interesting direction is not to treat the channel models as memory-less, i.e., as parallel BSCs but rather sequential. As the groups overlap, the test vector will indeed have dependent entries and, information-theoretically, one loses information by treating dependent entries as independent. However, we believe that incorporating memory to the decoder increases its computation overhead and most importantly, makes the solution not general and far from a real-world application. It is important to note that modeling the noise with a simple BSC provides the decoder the smallest possible information, but for the sake of practicality and generalization, we follow this simple model.

---

> > ### Author Response · Authors · 2023-11-20
> > **Response 2 to Reviewer CiYQ**
> >
> > Further, we would like to emphasize that $\Lambda$, similar to $\Delta'$, can be interpreted as a parameter that trades off between false alarm and misdetection. Indeed, $\Lambda=-\infty$ corresponds to always keeping all clients, $\Lambda=\infty$ discards all clients, and $\Lambda=0$ allows decisions to be made according to the prior belief, i.e., $\tilde{\delta}$. Hence, optimizing over $\Lambda$ is not fundamentally different from optimizing over $\Delta'$.
> >
> > - **Benchmark comparison**: As the reviewer points out, FedGT is inferior to RFA, given the top-1 accuracy as a benchmark. However, FedGT offers capabilities that RFA does not, e.g., identification capabilities. In the case of unintended data poisoning, this capability provides the opportunity to notify the client and clean the data, resulting in improved performance once completed. Furthermore, FedGT is only applied in a single round in the paper. With this in mind, we argue that FedGT, as presented in the paper, constitutes one of its weakest instantiations. However, it is still competitive with RFA---a scheme that is executed in every round of the training and does not offer the ability to identify mischief.
> >
> >     From the above, the challenge in comparing two defensive mechanisms as apples-to-apples is evident. Moreover, the reviewer is indeed correct in that RFA provides a secure aggregation mechanism, including all clients, whereas our instantiation of FedGT sacrifices some privacy to allow for identification. As there are no other schemes allowing for a reduction the privacy (except for the extreme of completely ignoring the privacy aspect), this is the closest to an apple-to-apples comparison we can achieve.
> >
> >     The RFA protocol securely computes the geometric median among all clients. Hence, as the reviewer points out, the RFA protocol performs secure aggregation on all of the clients. Although this provides more privacy than FedGT, we would like to emphasize that our scheme allows for identification and may be run in a single round. Hence, although RFA is sometimes superior in terms of top-1 accuracy, FedGT offers the possibility to identify malicious nodes. Therefore, in the case of unintended poisoning, poisoned nodes may be notified to look over their dataset. This has the potential of further improving the performance of the global model.
> >
> > - **Runtime concerns**: We acknowledge that the experiments consider a low number of clients compared to the maximum according to our definition. As far as the authors are aware, practical instantiations of cross-silo FL have so-far involved client constellations of size $10$, e.g., [1]. Hence, we chose the number of clients to be close to the numbers observed in practice.
> >
> >     Regarding the scaling, the main bottleneck is the decoding operation that has a complexity of $\mathcal{O}(2^m)$. Hence, as the number of tests grows, i.e., the number of rows in the assignment matrix, the decoder eventually becomes a bottleneck. Hence, provided a reasonable number of tests, FedGT scales to an arbitrary number of clients. Increasing the number of tests for a fixed number of clients will allow the server to obtain more fine-grained information about the clients at the expense of reduced privacy and a more complex decoding operation.
> >
> >     [1] - [https://www.melloddy.eu/](https://www.melloddy.eu/)

---

> > > ### Author Response · Authors · 2023-11-20
> > > **Response 3 to Reviewer CiYQ**
> > >
> > > - **Impact of mismatch**: First, the deteriorate effect of FA and MD depends on the context. An FA corresponds to a benign client being discarded. In a heterogeneous setting where clients possess unique datasets, discarding a benign client may have severe consequences as the global model will no longer be exposed to the unique data. For similar reasons, an MD may also have severe consequences under heterogeneity. On the other hand, in the homogeneous case, an FA may not impact performance notably as similar data resides among other clients. An MD, however, may still have a large impact on the performance depending on the metric of interest and the type of attack.
> > >
> > >     To demonstrate the tradeoff between false alarm and misdetection, we have rerun experiments for CIFAR10 using $\tilde{\delta}=0.1$ and $p=0.05$ for $\Lambda\in[-1,1]$ and $n_m\in \lbrace 1,3,5 \rbrace$. As can be seen in the table, for $\Lambda = 1.0$, we achieve an attack accuracy of $3\\%$, $4.1\\%$, and $7.6\\%$ (rounded in the tables) for $n_{m}\in\lbrace 1,3,5\rbrace$, respectively. Furthermore, for small $n_m$, the attack accuracy does not vary significantly with $\Lambda$. The top-1 accuracy is mainly unaffected by $\Lambda$ as the label-flip attack does not aim to reduce the top-1 accuracy.
> > >
> > >
> > >     Table for $n_m = 1$
> > >
> > >     | $\Lambda$ | $P_{FA}$ | $P_{MD}$ | ACC | ATT |
> > >     | --- | --- | --- | --- | --- |
> > >     | -0.8 | 0.0 | 0.0 | 0.83 | 0.04 |
> > >     | -0.6 | 0.0 | 0.0 | 0.83 | 0.04 |
> > >     | -0.4 | 0.0 | 0.0 | 0.83 | 0.04 |
> > >     | -0.2 | 0.0 | 0.0 | 0.83 | 0.04 |
> > >     | -0.0 | 0.0 | 0.0 | 0.83 | 0.04 |
> > >     | 0.2 | 0.0 | 0.0 | 0.83 | 0.04 |
> > >     | 0.4 | 0.07 | 0.0 | 0.83 | 0.04 |
> > >     | 0.6 | 0.07 | 0.0 | 0.83 | 0.04 |
> > >     | 0.8 | 0.07 | 0.0 | 0.83 | 0.04 |
> > >     | 1.0 | 0.14 | 0.0 | 0.83 | 0.03 |
> > >
> > >     Table for $n_m = 3$
> > >
> > >      | $\Lambda$ | $P_{FA}$ | $P_{MD}$ | ACC | ATT |
> > >     | --- | --- | --- | --- | --- |
> > >     | -0.8 | 0.0 | 1.0 | 0.83 | 0.12 |
> > >     | -0.6 | 0.08 | 0.67 | 0.83 | 0.11 |
> > >     | -0.4 | 0.08 | 0.67 | 0.83 | 0.11 |
> > >     | -0.2 | 0.08 | 0.67 | 0.83 | 0.11 |
> > >     | -0.0 | 0.08 | 0.67 | 0.83 | 0.11 |
> > >     | 0.2 | 0.08 | 0.33 | 0.83 | 0.06 |
> > >     | 0.4 | 0.08 | 0.33 | 0.83 | 0.06 |
> > >     | 0.6 | 0.08 | 0.33 | 0.83 | 0.06 |
> > >     | 0.8 | 0.17 | 0.33 | 0.82 | 0.07 |
> > >     | 1.0 | 0.17 | 0.0 | 0.82 | 0.04 |
> > >
> > >     Table for $n_m = 5$
> > >
> > >     | $\Lambda$ | $P_{FA}$ | $P_{MD}$ | ACC | ATT |
> > >     | --- | --- | --- | --- | --- |
> > >     | -0.8 | 0.0 | 1.0 | 0.81 | 0.31 |
> > >     | -0.6 | 0.1 | 0.8 | 0.82 | 0.22 |
> > >     | -0.4 | 0.1 | 0.8 | 0.82 | 0.22 |
> > >     | -0.2 | 0.1 | 0.8 | 0.82 | 0.22 |
> > >     | -0.0 | 0.1 | 0.8 | 0.82 | 0.22 |
> > >     | 0.2 | 0.1 | 0.6 | 0.81 | 0.21 |
> > >     | 0.4 | 0.1 | 0.6 | 0.81 | 0.21 |
> > >     | 0.6 | 0.1 | 0.6 | 0.81 | 0.21 |
> > >     | 0.8 | 0.1 | 0.4 | 0.82 | 0.11 |
> > >     | 1.0 | 0.1 | 0.2 | 0.82 | 0.08 |
> > >
> > >     It is worthwhile mentioning that the results presented here are based on a single realization of malicious users. For $n_{\mathrm{m}}=5$, we have a single subgroup consisting of only benign users and are, therefore, able to achieve a low attack accuracy without hampering the top-1 accuracy. The results of tables 1-3 in Appendix A are based on the average performance over five realizations, one of which does not yield a subgroup with only benign clients. Consequently, our simple test will use a malicious group as reference and yield a large identification error. This is the main reason for the low top-1 accuracy in Table 2 compared to, e.g., RFA.
> > >
> > > - **Unusual behavior of "no defense" experiments with CIFAR-10 in Figure 4**: We believe this is a manifestation of the label-flip attack, which does not significantly alter the model parameters. Hence, before convergence, the benign client updates are implicitly weakening the label-flip attack as all the clients are equally contributing to the model, according to the FedAvg protocol.
> > >
> > > - **Adversarial model**: The scenario under consideration is outlined in Section 3 where it is stated that the paper focuses solely on data-poisoning attacks. Although it would be useful to have a discussion pertaining to why we make this assumption in the main paper, due to space limitations we are forced to leave this to the appendix.

---

> ### Comment · Reviewer_CiYQ · 2023-11-21
> **Response to Response to Reviewer CiYQ**
>
> - **Structure and novelty:** To be clear, I am not against keeping some or even many of the coding theory aspects of the paper in Section 5. What I am saying is that the current presentation of Section 5 is unfair towards [1], where they are cited as a tangentially related paper, when this paper's techniques are practically lifted directly from [1].  This alone is grounds for rejection, but I do not think that the authors do this on purpose (or at least hope so). That said, it is a precondition for acceptance to properly give credit to [1]. My other point is that the current version of Section 5 is, as acknowledged by the authors, just not enough background for most ML researchers, including me. I found reading [1] the solution for myself. In particular, just putting the forward-backward algorithm formula does not help an average ML researcher to understand the coding theory/group testing theory involved, and doesn't give a coding theory expert any valuable information either. It is up to the authors how they want to structure Section 5 provided the credit to [1] is given, but another option will be to put a lot of additional information in the appendix. To be blunt here, the reviewer is not sure if there are any good solutions to presenting all the needed background to an ML crowd, the reviewer is just giving suggestions that can hopefully help.
> - **Assumption of knowing the target:** In a poisoning scenario where a whole class is attacked, I find the reasoning provided by the authors useful. The authors should include it in their experiment and threat model discussion. That said this only applies in this scenario and I think this scenario is not that common in practice.
> - **Assumptions $Q$:** Thank you for the informative answer.
> - **The hyperparameter $\Lambda$:** I still think that the paper will benefit from more principled way of setting it given its claimed connection to $\Delta'$ which can be set principally.
> - **Benchmark comparison:** The reviewer appreciates the additional capabilities of FedGT. However, the authors themselves point to poisoning prevention under aggregation as their main application. As such the worse benchmark results remain very relevant to this paper. My concern that RFA is "hampered" compared to FedGT remains. I know that apples-to-apples comparison is hard but the results do not reflect the advantage of FedGT of being able to choose the trade-off parameter.
> -**Impact of mismatch:** I am thankful to the authors for the additional experiments. I think providing experiments on multiple realizations in the next revision of the paper will be beneficial.
>
> All in all, I am happy with the engagement from the authors, but several big issues remain. In particular, one is properly citing [1] and making the paper more accessible to the ML community, and the other one is the results in comparison to RFA. I simply think that the paper needs another major revision, both experiment-wise and writing-wise. I cannot recommend acceptance. I am open to continuing the discussion, however, for the purpose of improving the paper.

---

### Official Review · Reviewer_znCD · 2023-10-27

**Soundness:** 3 good
**Presentation:** 3 good
**Contribution:** 3 good
**Rating:** 6
**Confidence:** 2

**Summary:**

In this paper, the authors employ group testing to detect malicious clients in FL. In particular, they investigate both vanilla FedAvg and secure aggregation algorithms in different parameter settings of group sizes. They conduct a formal analysis of the effectiveness and privacy guarantees on the varying group sizes. In addition, the authors evaluate their testing method in several datasets in terms of efficiency, utility, and effectiveness.

**Strengths:**

1. Using group testing to detect malicious clients in FL seems new and effective.

2. The analysis and evaluation on the proposed method are comprehensive and thorough.

**Weaknesses:**

1. The cost (particularly the communication) of group testing seems quite high, scale with the number of groups.

2. The privacy consideration seems not sufficient. I am wondering if the model difference between two testing groups could reveal individual's updates. For example, if group A includes C1, C2, C3, and group includes C2, C3, then the difference of the two aggregated models could reveal the model updates of C1.

**Questions:**

See my comments above.

---

> ### Author Response · Authors · 2023-11-16
> **Response to Reviewer znCD**
>
> We thank the reviewer for their review and for the questions raised. We would like to thank the reviewer for finding our idea novel and effective. Below, you may find our response.
>
> 1. **Communication Cost:**
>    We understand and are aware of the fact that secure aggregation comes with a high communication cost. In our work, we presented results where we do secure aggregation across many small groups of clients in a single communication round.
>    In Appendix B, we leverage state-of-the-art protocols to estimate and the communication overhead of our proposed framework.
>    For the experiments in the main body, we would like to emphasize that, compared to RFA, the communication overhead of FedGT is lower or similar in all rounds, including when group testing is performed.
>
> 2. **Privacy:**
>    The reviewer raises a valid concern. However, this is completely captured by the concept of minimum distance of the dual code (the code having the assignment matrix as a generator matrix). For the particular case raised by the reviewer, the assignment matrix will look like this:
>
>    $$\\mathbf{A}_1 = \\begin{pmatrix}
>        1 & 1 & 1 & 0 & \\dots &0 \\\\
>        0 & 1 & 1 & 0 & \\dots &0 \\\\
>        \vdots & \vdots & \vdots & \vdots & \vdots & \vdots
>    \end{pmatrix}$$
>
>    Then, the code defined by $\mathbf{A}_1$ as a generator matrix will have in its codebook (rowspan) the vector $(1,0,0, \dots, 0)$. In our application, this means that by linear combinations, the server can get the model C1. However, in our experiments, we use an assignment matrix that defines a code of minimum Hamming distance $4$, this ensures that any linear combination performed by the server will result in an aggregation of at least $4$ client models.
>    Hence, for the codes proposed in the paper, privacy is preserved.

---

### Official Review · Reviewer_6XYo · 2023-10-30

**Soundness:** 3 good
**Presentation:** 3 good
**Contribution:** 2 fair
**Rating:** 6
**Confidence:** 4

**Summary:**

The manuscript proposes FedGT, a secure aggregation framework for detecting malicious clients in federated learning. FedGT classifies each client into several overlapping groups, enabling the server to predict the presence or absence of malicious clients within each group. This allows the removal of such clients from the training of the global model. By adjusting the group sizes, FedGT achieves a balance between security and privacy. In addition, the manuscript investigates targeted label flip attacks and untargeted label permutation attacks, comparing their performance with RFA and oracle methods across varying numbers of attackers. Experimental results demonstrate that FedGT effectively identifies malicious clients, reducing attack accuracy.

**Strengths:**

- This manuscript's focus on secure federated learning is crucial in federated learning studies. The manuscript is well-structured, presenting a clear and coherent flow of ideas, which enriches the readability and understanding of the content.
- The method presented in this paper is innovative, as it combines group testing and secure federated learning aggregation techniques. It treats the identification of malicious clients as a decoding process, where the defective vector is determined based on the known distribution matrix and test results.
- The paper evaluates the proposed FedGT method, focusing on its effectiveness in mitigating poison attacks and safeguarding client privacy.

**Weaknesses:**

- The article's limitation in considering scenarios with a fixed number of clients (15 and 31) raises concerns about the scalability and performance of FedGT as the number of clients increases. Further investigation is necessary to assess the method's effectiveness in larger-scale scenarios and address potential challenges related to the size of the assignment matrix.
- In a related work section, the authors mention that RFA performs less well than other robust federated aggregation schemes. So why not use these better performance schemes as comparison schemes for FedGT?
- In Figure 3, the top-1 accuracy of FedGT is inferior to RFA and Oracle when the number of attackers is small and only exceeds RFA when the number of malicious clients under the subgraph (c) of Figure 3 is 5, which challenges the validity of the method.
- The authors should consider a larger number of attackers. Especially when the number of clients is 31, the number of attackers is too small to prove the effectiveness of FedGT to resist the collusion of multiple attackers.
- FedGT has more assumptions than a typical federated learning security aggregation algorithm. Prior knowledge includes the number of malicious clients and additional test sets, which weaken the contribution made in the paper.

**Questions:**

- In Appendix E, the authors demonstrate that the privacy of the FedGT implementation is equivalent to that of the security aggregation scheme for r clients. However, it would be helpful to provide a more formal and detailed explanation of the specific definition of privacy.
- The choice of decoding method will affect the performance of FedGT. What is the basis or reason for choosing the Neyman-Pearson scheme? Does Neyman-Pearson's hypothesis accurately describe the problem of reality?
- In addition to the test set, FedGT uses an additional enhanced data set. However, the authors need to provide more clarity regarding the composition and role of this data set. Specifically, the authors should describe in more detail what the enhanced data set comprises and how it limits the application of FedGT.

---

> ### Author Response · Authors · 2023-11-17
> **Response to Reviewer 6XYo**
>
> We thank the reviewer for their efforts and for the questions and for raising awareness regarding potential weaknesses of the paper.
>
> * **Scalability**: We thank the reviewer for pointing out this important aspect.
>     We acknowledge that the experiments are targeting scenarios with a low number of clients.
>     This choice is inspired from practical instantiations of cross-silo FL which, so far, typically has involved client constellations of sizes around $10$, see e.g., [1].
>
>     Regarding the scaling: the main bottleneck lies in the optimal decoding operation that has a complexity of $\mathcal{O}(2^m)$. Hence, as the number of test grows, i.e., the number of rows in the assignment matrix, the decoding operation eventually becomes infeasible.
>     Hence, provided a reasonable number of tests, FedGT scales to an arbitrary number of clients.
>     Increasing the number of tests for a fixed number of clients will allow the server to obtain more fine-grained information about the clients at the expense of reduced privacy and a more complex decoding operation.
>
>
>
> * **Comparison with RFA**: The reason why we compare FedGT's performance to RFA is that RFA also aims at preserving the privacy of the clients' data.
>     Other robust aggregation schemes do not attempt to preserve privacy as they typically rely on the server having access to the clients' individual models, thus exposing the client to different forms of attack, e.g., data reconstruction attacks.
>     Hence, in an attempt to compare apples-to-apples, we settled for RFA.
>
>     In the setting of an untargeted attack (Figure 3), our experiments have rendered RFA superior for lower prevalences and we agree that lower prevalences are more likely than higher one, especially for a cross-silo FL scenario. However, the performance of FedGT is competitive with RFA and it provides identification of malicious clients.
>     In the case of an unintentional poisoning attack, FedGT offers the possibility to notify the flagged client and have it fix the data issue whereafter it may join the federation, enabling improved performance.
>     Robust aggregation schemes like RFA does not offer this possibility.
>
> * **Number of attackers for the setting $n=31$**: For the experiments involving $n=31$ clients, we consider $n_m= 6$ malicious clients.
>     This corresponds to $20\%$ of the clients being malicious, a number that, to the authors, does not seem to be unreasonably small.
>     The rationale of this experiment is to demonstrate that FedGT scales beyond the experiments provided in the main body that includes only $15$ clients.
>     Moreover, the malicious clients still "collude" as they attempt to deteriorate the learning in the same way.
>     However, compared to the main body, each client has less data and therefore less power to influence the global model.
>     \item \textbf{Validation dataset at the server}: We would like to emphasize that the validation dataset utilized in our instantiation of FedGT is purely due to the testing strategies that rely on the group accuracy and the group recall, respectively.
>     The framework is valid for arbitrary testing strategies and others, not requiring a validation dataset, are possible.
>     One such test could rely on clustering the principal components of the aggregated subgroup models.
>     Furthermore, note that we assume a quasi-dataset for validation in the paper, i.e., a dataset distributed similarly---but not necessarily equal---to the clients' data.
>     Also, in our tests, we use a very small validation set consisting of $100$ data points, which is significantly smaller than the client datasets.

---

> ### Author Response · Authors · 2023-11-17
> **Review Response 2 to Reviewer 6XYo**
>
> * **Knowledge of the number of malicious clients**: As mentioned in the paper, we share this view with the reviewer.
>     Feeding the prevalence $\frac{n_m}{n}$ to the decoder requires perfect knowledge of $n_m$, something that is typically not available at the server.
>     However, we show in Appendix G that the decoder is robust to a mismatch in the prevalence, i.e., when an incorrect value $\tilde{\delta} \neq \delta$ is used instead. To strengthen this further, we performed simulations for a targeted attack on CIFAR-10 dataset, for $n_m \in \lbrace 1, 3, 5\rbrace$ where we only feed the decoder a mismatched value $\tilde{\delta} = 0.1$ and $p=0.05$.
>     Below, we present the results as the top-1 accuracy (ACC), attack accuracy (ATT), misdetection and false-alarm probabilities versus the decoding threshold $\Lambda$ for $n_m=1,2,3$, respectively.
>
>    $n_m = 1$
>     | $\Lambda$ | $P_{FA}$ | $P_{MD}$ | ACC | ATT |
>     | --- | --- | --- | --- | --- |
>     | -0.8 | 0.0 | 0.0 | 0.83 | 0.04 |
>     | -0.6 | 0.0 | 0.0 | 0.83 | 0.04 |
>     | -0.4 | 0.0 | 0.0 | 0.83 | 0.04 |
>     | -0.2 | 0.0 | 0.0 | 0.83 | 0.04 |
>     | -0.0 | 0.0 | 0.0 | 0.83 | 0.04 |
>     | 0.2 | 0.0 | 0.0 | 0.83 | 0.04 |
>     | 0.4 | 0.07 | 0.0 | 0.83 | 0.04 |
>     | 0.6 | 0.07 | 0.0 | 0.83 | 0.04 |
>     | 0.8 | 0.07 | 0.0 | 0.83 | 0.04 |
>     | 1.0 | 0.14 | 0.0 | 0.83 | 0.03 |
>
>    $n_m = 3$
>     | $\Lambda$ | $P_{FA}$ | $P_{MD}$ | ACC | ATT |
>     | --- | --- | --- | --- | --- |
>     | -0.8 | 0.0 | 1.0 | 0.83 | 0.12 |
>     | -0.6 | 0.08 | 0.67 | 0.83 | 0.11 |
>     | -0.4 | 0.08 | 0.67 | 0.83 | 0.11 |
>     | -0.2 | 0.08 | 0.67 | 0.83 | 0.11 |
>     | -0.0 | 0.08 | 0.67 | 0.83 | 0.11 |
>     | 0.2 | 0.08 | 0.33 | 0.83 | 0.06 |
>     | 0.4 | 0.08 | 0.33 | 0.83 | 0.06 |
>     | 0.6 | 0.08 | 0.33 | 0.83 | 0.06 |
>     | 0.8 | 0.17 | 0.33 | 0.82 | 0.07 |
>     | 1.0 | 0.17 | 0.0 | 0.82 | 0.04 |
>
>    $n_m = 5$:
>     | $\Lambda$ | $P_{FA}$ | $P_{MD}$ | ACC | ATT |
>     | --- | --- | --- | --- | --- |
>     | -0.8 | 0.0 | 1.0 | 0.81 | 0.31 |
>     | -0.6 | 0.1 | 0.8 | 0.82 | 0.22 |
>     | -0.4 | 0.1 | 0.8 | 0.82 | 0.22 |
>     | -0.2 | 0.1 | 0.8 | 0.82 | 0.22 |
>     | -0.0 | 0.1 | 0.8 | 0.82 | 0.22 |
>     | 0.2 | 0.1 | 0.6 | 0.81 | 0.21 |
>     | 0.4 | 0.1 | 0.6 | 0.81 | 0.21 |
>     | 0.6 | 0.1 | 0.6 | 0.81 | 0.21 |
>     | 0.8 | 0.1 | 0.4 | 0.82 | 0.11 |
>     | 1.0 | 0.1 | 0.2 | 0.82 | 0.08 |
>
>     From these experiments, we observe that the attack accuracy and accuracy are robust even when the prevalence $\tilde{\delta}$ and crossover probability $p$ are mismatched.
>
> * **Privacy definition**: We agree with the reviewer that the privacy investigation could be deepened.
>     To the best of our knowledge, analyzing the privacy of secure aggregation is still an open problem and subject to active research.
>     It is known that privacy increases with the number of clients being aggregated [2] but to the authors knowledge, no work is able to formally quantify the privacy of secure aggregation.
>     All in all, the reviewer's suggestion constitutes an interesting research direction, but it is beyond the scope of this paper.
>
>
>
> * **Neyman-Pearson decision criterion**: We thank the reviewer for the interesting question. We agree that the choice of decision criterion used is critical for the performance of FedGT. We describe briefly the main idea behind Neyman-Pearson decision criterion in Appendix~C and the rationale behind its usage.
>     The Neyman-Pearson decision criterion is optimal in the sense that it  minimizes one type of error (misdetection or false-alarm) subject to a constraint on the other type of error.
>     In machine learning applications, one typically targets a high true-positive rate and a low false-positive rate.
>     The true-positive rate is often measured at a fixed false-positive rate that is deemed acceptable.
>     Such a scenario directly translates into the Neyman-Pearson formulation.
>
> [1] - https://www.melloddy.eu/
>
> [2] - A. R. Elkordy, J. Zhang, Y. H. Ezzeldin, K. Psounis and S. Avestimehr, "How Much Privacy Does Federated Learning with Secure Aggregation Guarantee?", 2022.

---

> > ### Comment · Reviewer_6XYo · 2023-11-21
> >
> > Thanks to the authors for providing practical examples and reasons for the small-scale bottleneck in the rebuttal. I understand the author's efforts with limited resources and practical circumstances.
> >
> > Realistic cross-silo framework can support large-scale federated learning: I agree with the author's reference to a scale 10 in practical examples. Smaller scales may indeed exist in some specific scenarios. However, real-world cross-silo federated learning frameworks have demonstrated the ability to support large-scale clients. For example, some open-source frameworks, such as FATE and PySyft, have successfully supported the participation of hundreds or even thousands of clients in real-world applications. These frameworks improve the scalability and efficiency of the system by optimizing communication and computation strategies and adopting layering and aggregation techniques.
> > Some related work like [1] on cross-silo secure aggregation considers more than 100 clients: The number of clients considered in related research papers often exceeds 100, reflecting the focus of the academic community on large-scale cross-silo federated learning. A larger number of clients can provide a valuable reference for future practical applications.
> > The authors can give some reasonable optimization demos to solve the decoding problem caused by a large number of rows of the allocation matrix, which is a positive direction.
> >
> > I appreciate your explanation for comparing your method with RFA because RFA does not directly access the client models. However, please consider including a comparison with other robust aggregation approaches.
> >
> > Including a comparison with other robust aggregation methods would provide a more comprehensive evaluation of your proposed method and its effectiveness. This will enhance the credibility of your research and allow readers to better understand the strengths and weaknesses of different approaches.
> > It is encouraged that related works like [2-3] should listed that consider both robustness and privacy in the revised manuscript and compare the differences and advantages between them and your approach in detail.
> >
> > I understand the size of the validation set you stated when using the validation set and the reasons for its use. I endorse your use of a small additional validation set for group testing. I appreciate your explanation and the experimental data regarding the mismatch between prevalence and crossover probability. However, please provide more data highlighting larger differences between prevalence and crossover probability. This would further strengthen the evidence supporting the effectiveness of your proposed method in scenarios where there is a significant disparity between these two factors.
> >
> > Although there is no unified definition of privacy metrics for secure aggregation scenarios, it is not trivial to claim that a method satisfies privacy:
> > In the existing research, the privacy of federated learning has been proved from a theoretical point of view, for example, measured by differential privacy and mutual information. In addition, from a practical point of view, we can evaluate the privacy of methods by simulating privacy attacks. In the revised manuscript, please discuss the application of these theoretical measures in further detail and provide relevant theoretical proofs and analysis. This will help strengthen the reader's understanding of the privacy-preserving capabilities of your proposed method.
> > Although the intuition is that the fewer clients the server accesses, the better privacy, this is a trade-off between how strict and how loose the secure aggregation policy should be. It needs to be more rigorous to claim that the proposed approach protects privacy only from this perspective.
> >
> > Thanks for your acknowledgment of the significance of the decision criterion in the performance of FedGT. Your explanation of the Neyman-Pearson decision criterion in Appendix C and its usage rationale is helpful. Clarifying in the main text that the safe FL scenario is in good agreement with the Niemann-Pearson formulation can enhance the rigor and persuadability of the manuscript.
> >
> > [1] Cao, Xiaoyu, et al. "Fltrust: Byzantine-robust federated learning via trust bootstrapping." arXiv preprint arXiv:2012.13995 (2020).
> > [2] Xiang, Zihang, et al. "Practical Differentially Private and Byzantine-resilient Federated Learning." Proceedings of the ACM on Management of Data 1.2 (2023): 1-26.
> > [3] Dong, Caiqin, et al. "Privacy-Preserving and Byzantine-Robust Federated Learning." IEEE Transactions on Dependable and Secure Computing (2023).
> >
> > Overall, the paper has yet to make significant contributions, so I will not support the acceptance.

---

### Official Review · Reviewer_nACW · 2023-11-01

**Soundness:** 2 fair
**Presentation:** 3 good
**Contribution:** 2 fair
**Rating:** 3
**Confidence:** 3

**Summary:**

This paper proposes a method based on group testing to combine identification of malicious clients in FL with secure aggregation, which provides a tradeoff between privacy and Byzantine robustness.

**Strengths:**

1.	The ideal of using group testing to enable Byzantine resilience with secure aggregation is interesting.

**Weaknesses:**

1.	As the core technical contribution is to shift the paradigm from testing individuals to testing groups, and no novel testing strategy is proposed, the reviewer feels that this paper fits better venues on security or coding, compared with ML venues.

2.	More background and context need to be provided about error correcting codes and the design of assignment matrix A. For example, why one needs to choose A with good distance properties and how to choose it. Readers who are not familiar with coding theory may find it difficult to understand.

3.	Conventional linear block codes (defined by their generator matrices) are defined on finite fields, for which binary addition (or XOR) is used. However, in the context of group testing, the check node is evaluated using logical OR operation. It is not clear how conventional block codes can be directly used here as candidates for the assignment matrix.

**Questions:**

1.	In Figure 1(c), why FedGT outperforms oracle?

---

> ### Author Response · Authors · 2023-11-16
> **Response to Reviewer nACW**
>
> We thank the reviewer for the comments and the suggestions and for finding the proposed idea interesting. Regarding the weaknesses spotted in this review, we appreciate them and below you may find our comments.
>
> * We believe that the challenges of FL pertaining to privacy and security are important also for the ML community.
> We also believe that combining expertise from different scientific communities can benefit the development of resilient and privacy preserving FL.
> Furthermore, we think it is a good idea to present our framework in an ML venue to demonstrate that it is possible to identify malicious nodes, not only mitigating their impact.
> To the authors' knowledge, FedGT is the first method to achieve this and we believe it is an important cornerstone to provide privacy while securing FL from different attack techniques. In regards to novelty, we offer new insights into the privacy of sub-group testing by drawing on connections between coding theory and secure aggregation.
> Furthermore, our work is the first to demonstrate identification of malicious nodes in FL without inspecting individual client contributions.
> We have chosen one of the simplest testing strategies possible to demonstrate the inherent potential of leveraging group testing with FL even in the presence of very naive tests.
> We are currently working on extensions to more sophisticated tests.
>
> * We understand that it might be confusing for a potential reader who is not familiar with coding theory and we tried our best in Section 5 to introduce the relevant concepts.
> However, due to the page limits, we had to skip some details.
> Please do understand that coding theory is a research area that has been studied for more than 70 years, making it very challenging to incorporate all but the necessary details in a few pages.
> Furthermore, references are provided to the interested reader. The process of finding good codes and their corresponding parity-check matrix (used as an assignment matrix in the context of group testing) typically involves choosing a class of well-known codes (like BCH code, Reed-Muller codes). Usually, for short-length codes (up to ~100 bits), algebraic codes (BCH, Reed-Muller) are considered and the properties of these codes are well-studied.
> We follow this process in our paper and use a BCH code (length 15) for the experiments in the main body and a cyclic code (length 31) in Appendix J.
> In the submission, we cite the following book [1], where well-known (algebraic) codes are tabulated, but there also other works that provide a comprehensive list of channel codes.
>
>
>
> * As the reviewer points out, conventional binary channel codes are designed over the binary finite field. However, coding theory does not necessarily need to be based on the binary finite field. In fact, there exist also codes defined over rings rather than fields. However, we need to clarify that is not our contribution to use channel codes for group testing. Existing works [2] have showed that well-known channel codes help in designing nice pooling strategies.
>
>
>
> * Regarding the question of Figure 1(c), from the heterogeneous data among the clients, some clients, although benign, actually deteriorate the model by participating. Client 4, whose dataset consists mainly of examples labeled as class 2, see Fig. 13, constitutes such an example.  Our group testing framework, with the employed testing strategy, is tailored to reduce the attack accuracy in targeted attacks (and increase the balanced accuracy in untargeted attacks). Thus it may misclassify some clients that are deemed to deteriorate the global model.
> That is, from the group testing's point of view, client 4 seems to be compromised.
> Constructing defensive mechanisms that are able to handle such severe imbalance in label distribution is a challenging and open research problem.
>
> [1] - MacWilliams, F. J. (Florence Jessie), and Sloane, N. J. A. (Neil James Alexander), (1977). The theory of error correcting codes, Elsevier/North-Holland.
>
> [2] - A. Barg and A. Mazumdar, "Group Testing Schemes From Codes and Designs," in IEEE Transactions on Information Theory, vol. 63, no. 11, pp. 7131-7141, Nov. 2017.

---

> > ### Comment · Reviewer_nACW · 2023-11-22
> >
> > I appreciate the authors' efforts in resolving my concerns.
> >
> > However, I still feel that this work, as a direct application of group testing into secure aggregation, is a bit lacking in novelty, and its technical significance is a bit difficult for audiences in ML community to appreciate.
> >
> > Hence, I will stick to my previous rating.

---

### Meta-Review · Area_Chair_UQ9b · 2023-12-08

**Metareview:**

This paper proposes FedGT, a method for identifying malicious clients in FL with secure aggregation. Given clients separated into overlapping groups, FedGT applies group testing to identify the malicious client, which works even when client updates from the same group are aggregated. The authors demonstrate on datasets such as MNIST, CIFAR-10 and ISIC that FedGT can effectively mitigate against the label flipping attack.

Reviewers generally found the idea based on group testing to be novel and interesting. However, several critical weaknesses remain even after the rebuttal:
1. Cost of testing is very high, and the method does not scale well to a large number of clients.
2. Empirical performance is often inferior to simple robust aggregation techniques such as RFA. Other attack methods and defense baselines should be considered as well.
3. Beyond the idea of using group testing to detect malicious clients, there is relatively little technical contribution, especially for improving the cost of testing and its empirical performance. Reviewers also found the explanation of group testing to be unclear to an ML audience.

The authors are strongly encouraged to take reviewer feedback into consideration and address these weaknesses for the next revision.

**Justification For Why Not Higher Score:**

The method is not yet practical, and empirical evaluation is not comprehensive enough. Comparison against the RFA baseline (robust aggregation) is also not favorable.

**Justification For Why Not Lower Score:**

N/A

---

### Decision · Program_Chairs · 2024-01-16

Reject